# Peptide-based inflammation-responsive implant coating sequentially regulates bone regeneration to enhance interfacial osseointegration

Wei Zhou [1,2,5], Yang Liu [1,5], Xuan Nie [3,5], Chen Zhu [1] ✉, Liming Xiong [2] ✉, Jing Zhou [4] ✉ & Wei Huang [1] ✉

Aseptic loosening is the primary cause of bone prosthesis failure, commonly attributed to inadequate osseointegration due to coatings misaligned with bone regeneration. Here, we modify the titanium surface with a mussel-inspired peptide to form a 3,4-dihydroxyphenylalanine (DOPA)-rich coating, then graft $N_3$-K15-PVGLIG-K23 (P1) and $N_3$-Y5-PVGLIG-K23 (P2), which are composed of anti-inflammatory (K23), angiogenic (K15), osteogenic (Y5), and inflammation-responsive (PVGLIG) sequences, onto the surface via click chemistry, forming the DOPA-P1@P2 coating. DOPA-P1@P2 promotes bone regeneration through sequential regulation. In the initial stage, the outermost K23 induces M2 macrophage polarization, establishing a pro-regenerative immune microenvironment. Subsequently, K15 and Y5, exposed by the release of K23, enhance angiogenesis and osteogenesis. In the final stage, DOPA-P1@P2 outperforms the $TiO_2$ control, showing a 161% increase in maximal push-out force, a 207% increase in bone volume fraction, and a 1409% increase in bone-to-implant contact. These findings show that DOPA-P1@P2 efficiently enhances interfacial osseointegration by sequentially regulating bone regeneration, providing viable insights into coating design.

Each year, more than one million joint arthroplasty procedures are performed in orthopaedic practice worldwide, with the ultimate goals of alleviating pain and improving function[1]. Despite the clinical success of bone prostheses, complications remain, which can result in both implant failure and the need for revision surgery. Aseptic loosening due to inadequate osseointegration at the bone–implant interface is a major complication, accounting for over half of implant failures[2]. In general, osseointegration includes the direct deposition of regenerated bone on the implant surface and the consequent formation of solid bone-to-implant anchors without fibrous interference, emphasizing the importance of satisfactory peri-implant bone regeneration in osseointegration[3]. In recent decades, titanium and its alloys have become the leading materials for orthopaedic implants because of their improved biocompatibility, mechanical properties, and corrosion resistance[4]. However, titanium-based implants are considered biologically inert and require surface modifications to provide

[1]Department of Orthopaedics, The First Affiliated Hospital of USTC, Division of Life Sciences and Medicine, University of Science and Technology of China, Hefei, China. [2]Department of Orthopaedics, Union Hospital, Tongji Medical College, Huazhong University of Science and Technology, Wuhan, China. [3]Department of Pharmacy, The First Affiliated Hospital of USTC, Division of Life Sciences and Medicine, Anhui Provincial Key Laboratory of Precision Pharmaceutical Preparations and Clinical Pharmacy, University of Science and Technology of China, Hefei, China. [4]Department of Urology, The First Affiliated Hospital of USTC, Division of Life Sciences and Medicine, University of Science and Technology of China, Hefei, China. [5]These authors contributed equally: Wei Zhou, Yang Liu, Xuan Nie. ✉e-mail: zhuchena@ustc.edu.cn; xiongliming@hust.edu.cn; jzhou21@ustc.edu.cn; zgkdhwei@ustc.edu.cn

sufficient bioactivity for osseointegration[5]. Traditional surface modifications often involve tuning hierarchical structures and extracellular matrix (ECM) functions *via* physiochemical or biomimetic approaches to promote osteogenesis. However, the clinical efficacy of such implants has been impeded by the discrepancies between in vitro and in vivo studies[6]. Recently, extensive osteoimmunology research has demonstrated that titanium-based implant surfaces that are nanotopographically modified, have increased hydrophilicity or are modified with bioactive peptides can facilitate M2 macrophage polarization and promote peri-implant bone regeneration in vivo[7–9]. Although recent data suggest the potential for clinical application, enhanced interfacial osseointegration remains to be accomplished[10]. A comprehensive understanding of the mechanisms underlying bone regeneration will offer viable insights for the development of innovative bioactive surfaces, thereby ensuring improved implant osseointegration.

Implant-mediated bone regeneration is a multifaceted physiological process that involves various cells and factors and comprises three stages, inflammation, proliferation and remodeling, over consecutive timeframes[11,12]. During the inflammation stage, a hematoma rapidly forms surrounding the implant surface and triggers the release of inflammatory mediators. Macrophages recruited to the implant site produce matrix metalloproteinases (MMPs) and degrade the ECM to facilitate bone reconstruction[13]. In response to microenvironmental signals, they undergo distinct M1–M2 polarization[14,15]. M1 macrophages secrete high levels of inducible nitric oxide synthase (iNOS) and proinflammatory cytokines, including tumor necrosis factor-alpha (TNF-α) and interleukin-1 beta (IL-1β)[16]. Prolonged M1 polarization induces transformation of acute inflammation into a chronic state with the subsequent formation of peri-implant fibrous capsule leading to implant failures[17]. Conversely, M2 macrophages express high levels of anti-inflammatory markers such as cluster of differentiation 206 (CD206), arginase-1 (Arg-1) and interleukin-10 (IL-10)[18]. Increased M2 macrophage polarization promotes the transition from the inflammation stage to the proliferation stage[19]. During the proliferation stage, neovascularization serves as a crucial connector between the bone and surrounding tissues to facilitate the transport of oxygen, nutrients, and bone marrow stromal cells (BMSCs) to the implant site[20]. Moreover, the local release of vascular endothelial growth factor (VEGF) and osteogenic growth peptide (OGP) promotes angiogenesis and BMSC osteogenic differentiation, respectively[21,22]. Once the inflammation and proliferation stages progress properly, the remodeling stage will follow smoothly to complete bone regeneration. A thorough understanding of the physiological process of bone regeneration provides valuable insights to improve osseointegration, as the implant surface should be functionally designed to align with the temporal sequence of the bone regeneration process.

In this work, we develop an inflammation-responsive multifunctional coating using four peptides that correspond to the sequential targets of the described stages: (1) K23 (KAFAKLAARLYRKALARQLGVAA), an anti-inflammatory peptide for immune regulation during the inflammation stage[23]; (2) PVGLIG, an MMP-2/9-cleavable peptide for application during the inflammation stage[24]; (3) K15 (KLTWQELYQLKYKGI), a VEGF-mimetic peptide for angiogenesis promotion during the proliferation stage[25]; and (4) Y5 (YGFGG), an OGP-derived peptide for osteogenesis promotion during the proliferation stage[26] (Fig. 1A–C). We assemble these peptides to construct a peptide-coated titanium-based coating (DOPA-P1@P2) by combining a mussel-inspired biomimetic strategy with bioorthogonal click chemistry techniques. Upon implantation, the outermost K23 layer modulates acute inflammation at the first stage of bone regeneration, facilitating M1 to M2 macrophage polarization. Moreover, the MMP-2/9 released by activated macrophages cleaves the PVGLIG sequence, resulting in detachment of the K23 layer and exposure of the underlying K15 and Y5 layers to promote angiogenesis and osteogenesis at the second stage and subsequent bone remodeling (Fig. 1F–H). In summary,

DOPA-P1@P2 effectively promotes bone regeneration by intentionally aligning with the physiological process of bone regeneraiton, thereby enhancing osseointegration at the bone–implant interface through sequential regulation.

## Results

### Titanium surface modification and material characterization

To engineer an inflammation-responsive biomimetic titanium surface that can orchestrate bone regeneration, we initially synthesized a mussel-inspired peptide ((DOPA)$_4$-OEG$_5$-DBCO) bearing a clickable dibenzocyclooctyne (DBCO) group via solid-phase peptide synthesis (Fig. 2A and Supplementary Fig. 1D). In this work, we employed readily available Fmoc-DOPA (acetone)-OH to incorporate DOPA into the peptide chain. To ensure that the mussel-inspired peptide adheres to titanium surfaces and allows subsequent bioorthogonal click reactions, we separated the tetrameric DOPA structures with glycine (Gly), lysine (Lys), and OEG-linked DBCO, resulting in the clickable biomimetic peptide Ac-(DOPA)-Gly-(DOPA)-Lys[(OEG$_5$)-(DBCO-COOH)]-(DOPA)-Gly-(DOPA) (Supplementary Fig. 1A). The chemical structure of (DOPA)$_4$-OEG$_5$-DBCO was analyzed via proton nuclear magnetic resonance ($^1$H NMR) spectroscopy, confirming the presence of a DOPA unit by identifying a distinctive peak at 8.64 ppm attributed to the phenolic hydrogens of the catechol group (Fig. 2J). To meet the functional demands of various stages of bone regeneration, four types of peptides are introduced for surface modification: (1) the anti-inflammatory sequence K23, KAFAKLAARLYRKALARQLGVAA; (2) the angiogenic sequence K15, KLTWQELYQLKYKGI; (3) the osteogenic sequence Y5, YGFGG; and (4) the inflammatory response sequence, PVGLIG. The inflammatory response sequence PVGLIG was placed between K23 and K15, as well as between K23 and Y5, to generate two composite peptide sequences that were subsequently modified with two-azidoacetic acid (-N3). This process yielded two azide-modified composite functional peptides capable of inducing an inflammatory response: P1 (N$_3$-K15-PVGLIG-K23) and P2 (N$_3$-Y5-PVGLIG-K23) (Fig. 2B–C, Supplementary Fig. 1B–C and 2A–B).

The synthetic mussel-inspired peptides, P1 and P2 were analysed via high-performance liquid chromatography (HPLC), confirming that both peptides exhibited high purities exceeding 96.6% (Fig. 2D–F). Electrospray ionization mass spectrometry (ESI-MS) evaluated the molecular masses of these peptides. The findings suggested that the monoisotopic masses [M + 2H]$^{2+}$ of (DOPA)$_4$-OEG$_5$-DBCO, [M + 6H]$^{6+}$ of P1, and [M + 5H]$^{5+}$ of P2 were 799.82, 834.32, and 718.65 Da, respectively; these findings are in alignment with the theoretical molecular masses of the peptides (1597.67, 5000.03, and 3588.28 Da, respectively) (Fig. 2G–I). These findings verified the successful production of mussel-inspired peptides and two composite functional peptides. An inflammation-responsive biomimetic titanium surface was then developed in two key steps. Initially, titanium plates or rods were submerged in a 0.01 mg/mL (DOPA)$_4$-OEG$_5$-DBCO solution, enabling coating with mussel-inspired peptides. In the second step, these peptide-coated plates or rods were separately immersed in 0.1 mg/mL solutions of P1, P2, or a 1:1 mixture of both, enabling the creation of a variety of modified surfaces through bioorthogonal click chemistry reactions between -DBCO and -N$_3$ (Fig. 1D–E). A 1:1 ratio of P1 to P2 was chosen on the basis of preliminary experimental results (Supplementary Fig. 3A–G), as this ratio enabled the modified surface to exhibit both angiogenic and osteogenic capabilities. Plates or rods immersed in PBS were designated the TiO$_2$ group and served as the control. Plates or rods immersed only in the mussel-inspired peptide solution were classified as the DOPA group, whereas those further grafted with P1, P2, or their mixture were classified as the DOPA-P1, DOPA-P2, and DOPA-P1@P2 groups, respectively.

Atomic force microscopy (AFM) was employed to assess the topography across various surfaces, revealing a significant increase in the roughness of surfaces modified with peptides compared with that

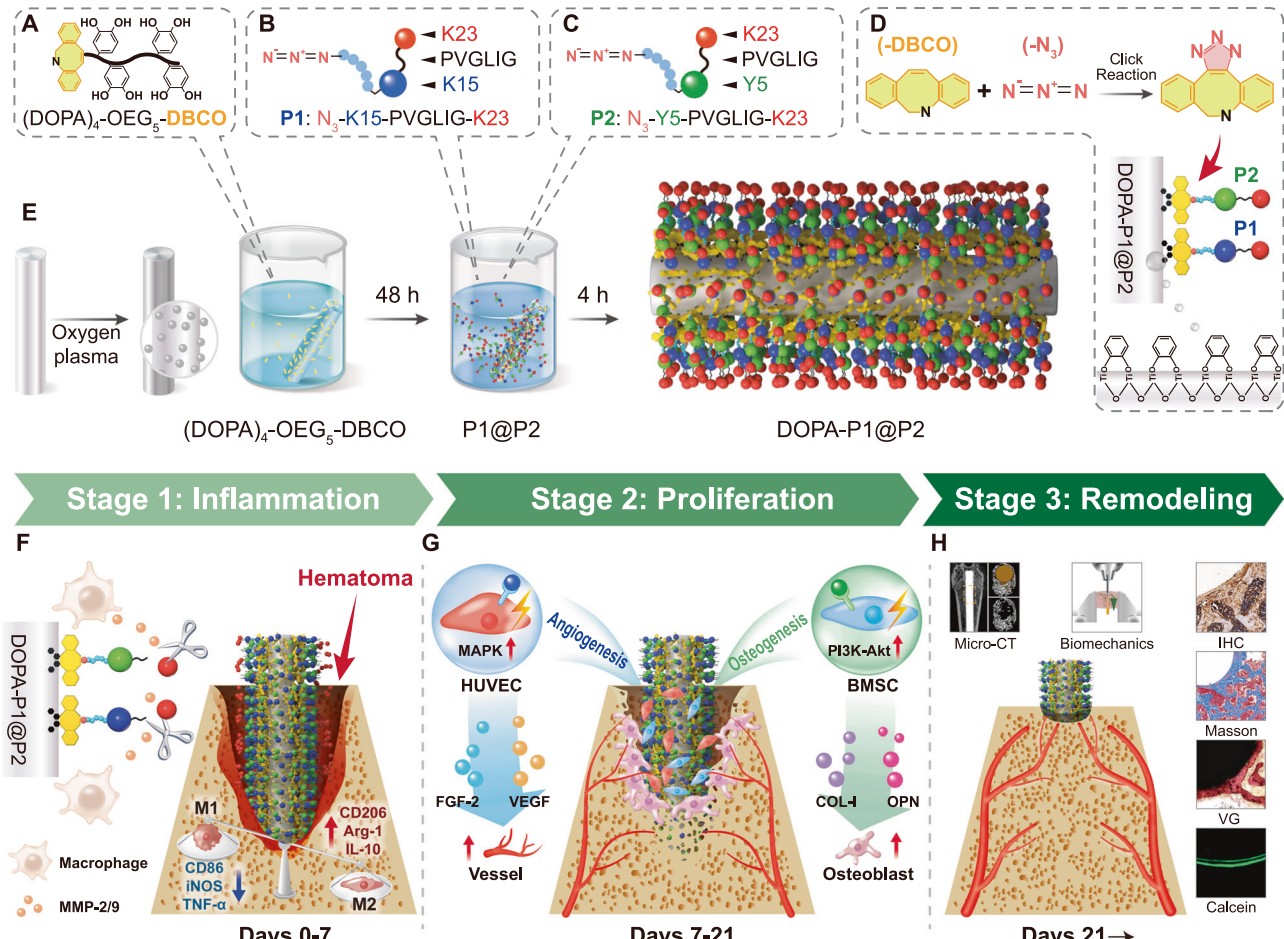

**Fig. 1 | The synthesis of DOPA-P1@P2 and its promotion of osseointegration via temporal regulation of bone regeneration. A** Mussel-inspired peptide modified with a bioclickable DBCO group: (DOPA)$_4$-OEG$_5$-DBCO. **B** Azide-modified multi-functional peptide P1 with angiogenic and anti-inflammatory properties: N$_3$-K15-PVGLIG-K23. **C** Azide-modified multifunctional peptide P2 with osteogenic and anti-inflammatory properties: N$_3$-Y5-PVGLIG-K23. **D** Mussel-inspired peptides bind to titanium surfaces through spontaneous coordination between catechol groups and titanium ions, while P1 and P2 are grafted onto biomimetic titanium surfaces via a bioorthogonal click chemistry reaction between N$_3$ and DBCO. **E** The preparation process of the biomimetic coating (DOPA-P1@P2). **F** DOPA-P1@P2 promotes M2 macrophage polarization during the first stage of bone regeneration. **G** DOPA-P1@P2 promotes angiogenesis and osteogenesis during the second stage of bone regeneration. **H** In the third stage of bone regeneration, the implant interface achieves satisfactory osseointegration.

of the TiO$_2$ surfaces (Fig. 2K, L). Interestingly, additional grafting of P1 or P2 onto DOPA-modified surfaces further enhances their roughness, likely due to variations in the peptide chain length or conformation. Subsequent analysis using static water contact angle measurements to evaluate the hydrophilicity across various surfaces, showed that the peptide-modified surfaces exhibited significantly lower water contact angles compared to the TiO$_2$ group (Fig. 2M, N). More precisely, the DOPA-P1@P2 group exhibited a significantly lower water contact angle (55.6°) compared to TiO$_2$ group (81.1°), indicating improved surface hydrophilicity. The compositions of various modified surface elements were characterized via X-ray photoelectron spectroscopy (XPS). As illustrated in Fig. 2O–P, surfaces modified with mussel-inspired peptides presented a notable increase in the N 1$s$ signal relative to that of the TiO$_2$ group, with additional grafting of P1, P2, or their mixture further amplifying this signal, consistent with the amino content on these surfaces. To confirm that the peptides grafted onto the titanium surfaces remained stable and retained long-term activity, the DOPA-P1@P2 surfaces were incubated in Dulbecco's modified Eagle's medium (DMEM) at 37 °C for three weeks prior to XPS analysis. The results revealed that the N 1$s$ signal decreased by less than 15% after three weeks (Fig. 2Q), indicating that the modified surface retained long-term biological activity. In this work, we utilized MMP-2 to simulate the local microenvironment during the inflammatory phase of bone regeneration and investigated the responsive release trends of an anti-inflammatory peptide (K23) from surfaces P1 and P2. The findings indicated that treatment with MMP-2 markedly enhances the release of K23 due to the specific enzymatic cleavage of the PVGLIG peptide sequence. Specifically, within the initial 24 h after MMP-2 treatment, the release rate of K23 exceeded 60%; the release rate then significantly slowed after day 3, and by the end of week 1, the release rate was approximately 85% (Fig. 2R). These findings demonstrate that modified titanium surfaces release substantial amounts of anti-inflammatory peptides early in inflammation to modulate immune responses, with K23 nearly depleted within 1 week. During the process of K23 release, its underlying angiogenic peptide sequence (K15) and osteogenic peptide sequence (Y5) become progressively exposed, thereby contributing to the second stage of bone regeneration. In summary, we effectively developed an inflammation-responsive biomimetic titanium surface (DOPA-P1@P2), which holds potential for the sequential regulation of bone regeneration.

## Evaluation of biocompatibility
In this study, bone marrow-derived macrophages (BMDMs), human umbilical vein endothelial cells (HUVECs), and bone marrow-derived

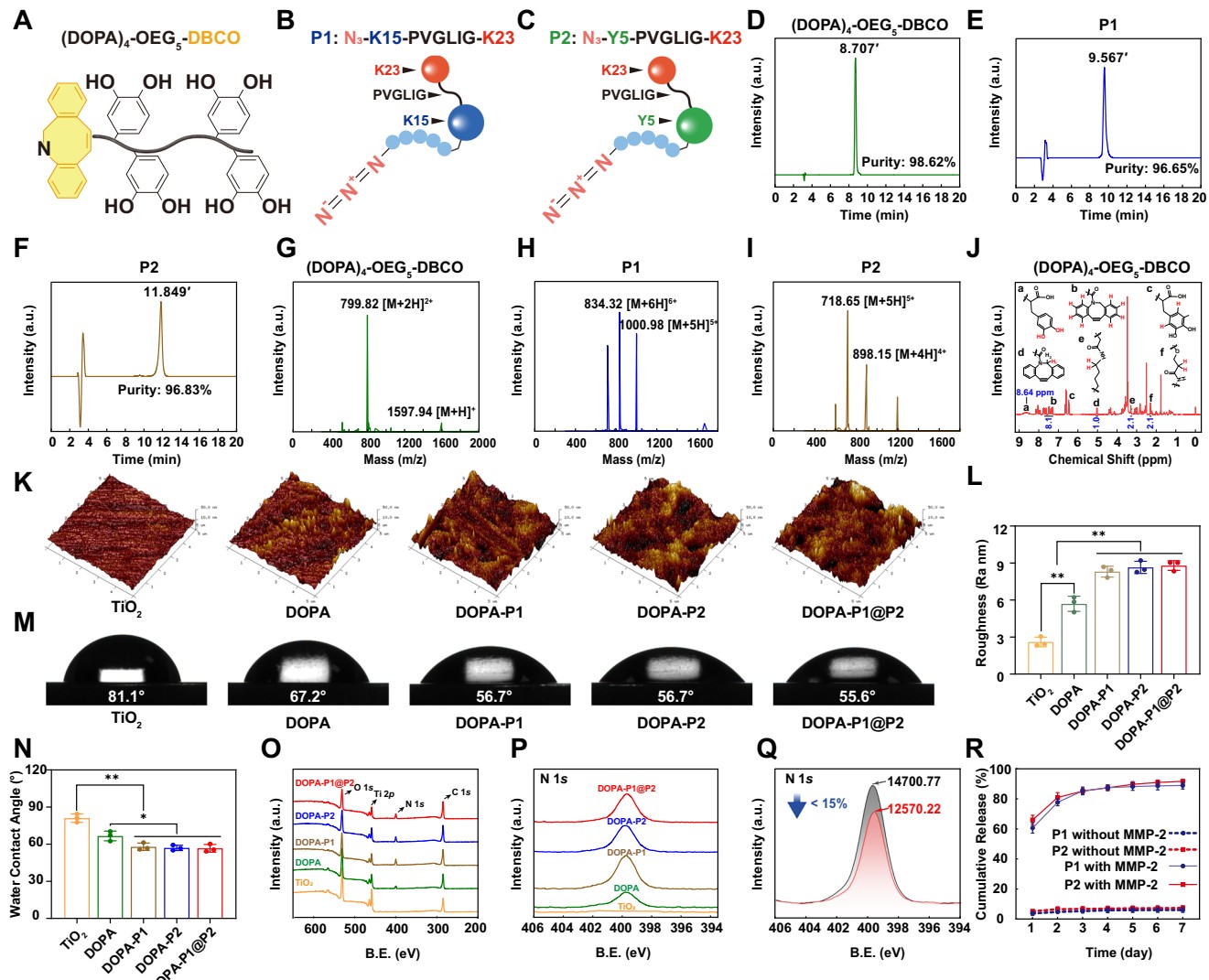

**Fig. 2 | Material characterization of modified surfaces. A–C** Schematic representations of the (DOPA)$_4$-OEG$_5$-DBCO, P1 and P2 structures. **D–F** The HPLC spectra of (DOPA)$_4$-OEG$_5$-DBCO, P1, and P2 show purities exceeding 96%. **G–I** ESI-MS spectra of (DOPA)$_4$-OEG$_5$-DBCO, P1, and P2. **J** The $^1$H NMR spectrum of (DOPA)$_4$-OEG$_5$-DBCO displays characteristic peaks (**a–f**) with corresponding area integrations highlighted in blue. The DOPA unit (**a**) is marked by a peak at 8.64 ppm. **K, L** AFM images and surface roughness quantification of modified surfaces ($n = 3$ per group). **M, N** Water contact angles and corresponding quantitative results for modified surfaces ($n = 3$ per group). **O** Broad spectrum analysis of XPS for modified surfaces. **P** N 1 $s$ detailed spectral analysis of XPS for modified surfaces. **Q** Changes in the N 1 $s$ signal in the XPS spectrum of DOPA-P1@P2 after three weeks of incubation in DMEM. **R** Response and release trends of P1 and P2 following treatment with MMP-2 solution ($n = 3$ per group). The data are presented as the mean ± standard deviation (SD). Data were analyzed by one-way ANOVA with Tukey's post hoc test, and *$p < 0.05$ and **$p < 0.01$ indicate statistical significance. Exact $p$ values were given in the Source Data file.

mesenchymal stem cells (BMSCs) were used as research subjects to evaluate cytocompatibility of DOPA-P1@P2 across different cell types. Initially, live/dead staining experiments were performed, revealing that all three cell types exhibited robust survival on the various modified surfaces. Furthermore, there were no notable differences in the ratios of live to dead cells across the five groups: TiO$_2$, DOPA, DOPA-P1, DOPA-P2, and DOPA-P1@P2 (Fig. 3A–C). We subsequently quantified the secretion of lactic dehydrogenase (LDH) from the aforementioned three cell types seeded on various modified surfaces to assess their cytotoxicity. Following a 24-h incubation period, the detected levels of LDH did not significantly differ among the groups, indicating that the peptide-modified surfaces exhibited no toxicity towards the cells (Fig. 3E–G). Additionally, the absorbance at 450 nm for three different cell types on various modified surfaces was measured at 24 and 72 h using CCK-8 assays (Fig. 3H–M). The findings indicated that, by day 3, the absorbance at 450 nm on the four modified surfaces (DOPA, DOPA-

P1, DOPA-P2, and DOPA-P1@P2) was higher compared to the TiO$_2$ control surface, suggesting that peptide modifications of titanium surfaces significantly promote cell proliferation. Finally, after a 12-h cultivation period, the morphological features of BMSCs on various substrates were analyzed via cytoskeletal staining, employing fluorescein isothiocyanate (FITC)-phalloidin and 4′,6-diamidino-2-phenylindole (DAPI). The findings demonstrated that on TiO$_2$ surfaces, BMSCs mostly displayed a spherical shape with scant filopodia, while on peptide-modified surfaces, the cells adopted polygonal forms and showed increased expression of filamentous actin (F-actin), particularly prominent in the DOPA-P1@P2 group (Fig. 3D). The materials characterization results suggest that the DOPA-P1@P2 surfaces, compared to the TiO$_2$ group, exhibit significantly increased roughness and hydrophilicity (Fig. 2K–N), which may explain the enhanced adhesion and spreading of BMSCs. Besides in vitro assays, we performed in vivo validation experiments in Sprague–Dawley (SD) rats. We implanted

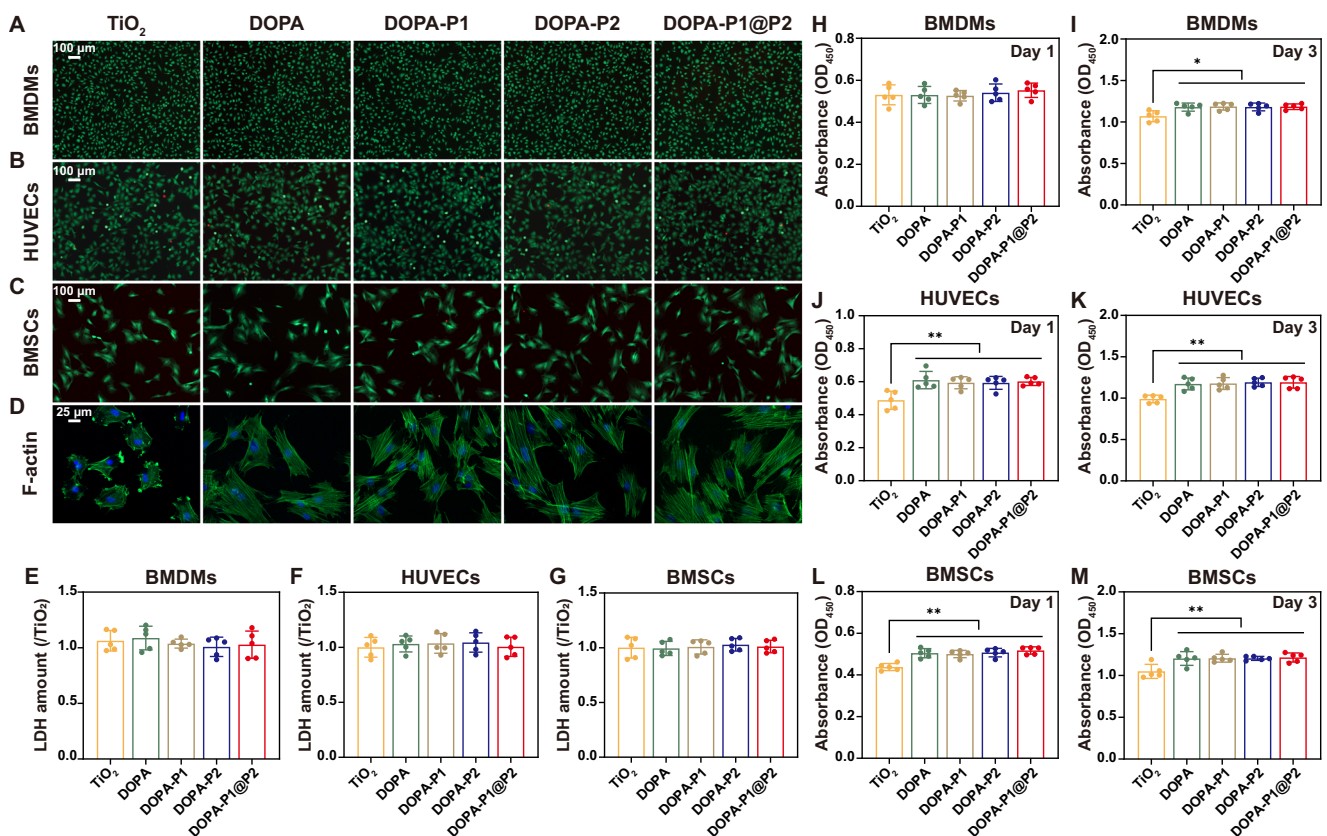

**Fig. 3 | Cytocompatibility assessment of modified surfaces. A–C** Live/dead staining of BMDMs, HUVECs, and BMSCs on modified surfaces. **D** Cytoskeletal staining of BMSCs on modified surfaces using FITC-phalloidin and DAPI. **E–G** LDH cytotoxicity assays for BMDMs, HUVECs, and BMSCs cultured on modified surfaces ($n = 5$ per group). **H–M** CCK-8 assays for BMDMs, HUVECs, and BMSCs cultured on modified surfaces over 1 and 3 days ($n = 5$ per group). The data are presented as the mean ± SD. Data were analyzed by one-way ANOVA with Tukey's post hoc test, and \*$p < 0.05$ and \*\*$p < 0.01$ indicate statistical significance. Exact $p$ values were given in the Source Data file.

modified titanium rods into rat femurs, and after two months, organs were analysed via haematoxylin and eosin (H&E) staining. The results confirmed that all the modified surfaces were free from visceral toxicity (Supplementary Fig. 4). In brief, the findings derived from in vitro and in vivo studies suggest that the developed peptide-coated surfaces enhance cellular adhesion and proliferation and are nontoxic to organisms, establishing the necessary foundation for regulating bone regeneration.

### Evaluation of immunomodulatory effects

To determine whether the inflammation-responsive biomimetic modified surfaces can effectively modulate immune responses, we cultured BMDMs on various groups of titanium surfaces, then conducted immunofluorescence staining. The results revealed that surfaces devoid of anti-inflammatory peptide modifications (TiO2 and DOPA) presented a marked increase in the expression of M1 macrophage markers (iNOS, depicted in purple), whereas surfaces modified with anti-inflammatory peptides (DOPA-P1, DOPA-P2, and DOPA-P1@P2) showed a marked upregulation of M2 macrophage markers (CD206, depicted in red) (Fig. 4A–C). These findings suggest that the DOPA-P1, DOPA-P2, and DOPA-P1@P2 surfaces are capable of effectively regulating immune responses and facilitating macrophage polarization to the M2 phenotype. The proportion of CD86⁺ cells (M1) was notably higher in the TiO2 and DOPA groups compared to the other groups, as revealed by flow cytometry analysis (Fig. 4F–G). Conversely, the proportion of CD206⁺ cells (M2) was significantly higher in the DOPA-P1, DOPA-P2, and DOPA-P1@P2 groups than in the TiO2 and DOPA groups (Fig. 4F, H). The western blot findings were

consistent with the aforementioned findings, revealing high expression of an M1-associated protein (iNOS) in the TiO2 and DOPA groups, whereas an M2-associated protein (CD206) was highly expressed in the DOPA-P1, DOPA-P2, and DOPA-P1@P2 groups (Fig. 4I–K). Moreover, the enzyme-linked immunosorbent assay (ELISA) findings demonstrated a substantial elevation in the concentrations of TNF-α within the TiO2 and DOPA groups compared with the DOPA-P1, DOPA-P2, and DOPA-P1@P2 groups (Fig. 4D). Conversely, the concentrations of IL-10 were elevated in the DOPA-P1, DOPA-P2, and DOPA-P1@P2 groups (Fig. 4E). Additionally, we observed that the concentrations of angiogenesis-related protein (VEGF) and osteogenesis-related protein (BMP-2) were significantly higher in the DOPA-P1, DOPA-P2, and DOPA-P1@P2 groups compared to the TiO2 and DOPA groups (Supplementary Fig. 5A–B). Furthermore, in addition to protein-level analyses, analyses were also conducted at the gene level. The M1-associated gene (*Ccr-7*) was highly expressed in the TiO2 and DOPA groups, whereas the M2-associated gene (*Cd206*), along with the angiogenesis-related gene (*Vegf*) and osteogenesis-related gene (*Bmp2*), exhibited elevated expression in the DOPA-P1, DOPA-P2, and DOPA-P1@P2 groups, as confirmed by reverse transcription quantitative polymerase chain reaction (RT-qPCR) analysis (Fig. 4L–O). The combined results from in vitro experiments indicated that the modified surfaces engineered in this study effectively promote M2 polarization of macrophages, thereby enhancing angiogenesis and osteogenesis.

In order to assess the in vivo immunomodulatory effects, various treated titanium rods were inserted into the femurs of rats. Five days post-implantation, the femurs were extracted for histological analysis. The findings of H&E staining demonstrated a substantial decrease in

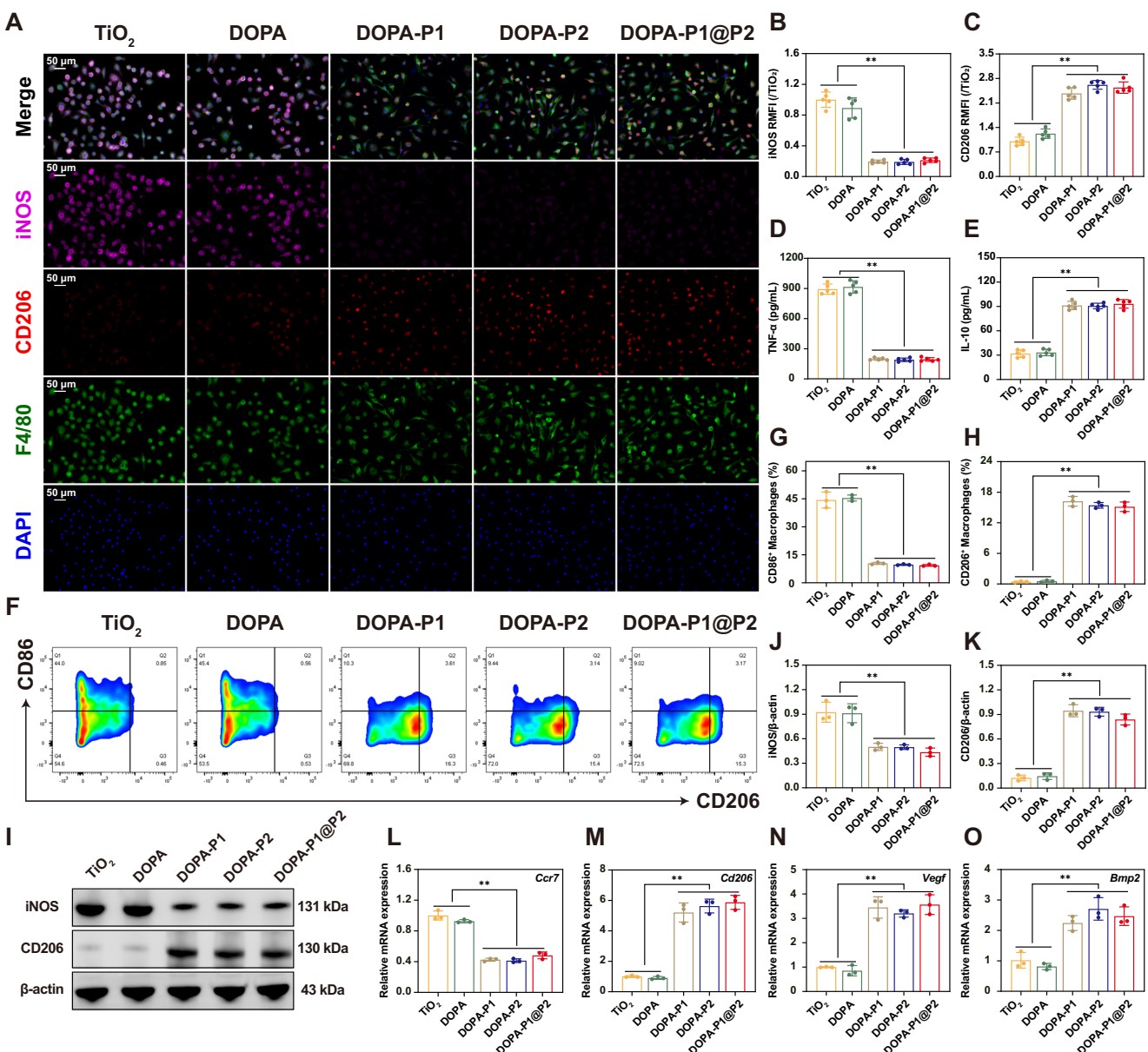

**Fig. 4 | Regulation of macrophage polarization in vitro. A** Immunofluorescence staining evaluated macrophage polarization in BMDMs cultured on various modified surfaces, with the macrophage marker F4/80 visualized in green, the M1 marker iNOS in purple, the M2 marker CD206 in red, and nuclei in blue. **B, C** Quantitative analysis of immunofluorescence staining for target proteins (n = 5 per group). **D, E** ELISA quantified the secretion levels of the proinflammatory cytokine TNF-α and the anti-inflammatory cytokine IL-10 (n = 5 per group). **F–H** Flow cytometry analyzed the expression of CD86 (M1 marker) and CD206 (M2

marker), and the corresponding quantitative analysis results (n = 3 per group). **I–K** WB detected the expression levels of iNOS and CD206 proteins and the corresponding quantitative analysis results (n = 3 per group). **L–O** RT-qPCR detected the gene expression levels of *Ccr7* (M1 marker), *Cd206* (M2 marker), *Vegf* (angiogenesis marker), and *Bmp2* (osteogenesis marker) (n = 3 per group). The data are presented as the mean ± SD. Data were analyzed by one-way ANOVA with Tukey's post hoc test, and *$p < 0.05$ and **$p < 0.01$ indicate statistical significance. Exact $p$ values were given in the Source Data file.

the fibrous layer thickness adjacent to the titanium rods in the DOPA-P1, DOPA-P2, and DOPA-P1@P2 groups compared to the TiO₂ and DOPA groups (Fig. 5A, F). Additionally, immunohistochemical staining demonstrated a substantial enlargement of the area showing positive IL-10 staining in the DOPA-P1, DOPA-P2, and DOPA-P1@P2 groups, whereas a significantly increase in the area positive for TNF-α staining was identified in the TiO₂ and DOPA groups (Fig. 5B, C, G, H). Moreover, immunofluorescence staining was performed to evaluate macrophage polarization phenotypes surrounding the titanium rods. The abundance of M1 macrophages (CD68⁺CD86⁺) was notably elevated in the TiO₂ and DOPA groups compared to the rest, whereas M2

macrophages (CD68⁺CD206⁺) were more abundant in the DOPA-P1, DOPA-P2, and DOPA-P1@P2 groups compared to the TiO₂ and DOPA groups (Fig. 5D, E, I, J). These results suggest that the developed surfaces modified with anti-inflammatory peptides can effectively modulate immunity during the phase of inflammation in vivo, promoting the shift from M1 to M2. In summary, data from in vitro and in vivo experiments indicated that the inflammation-responsive biomimetic titanium surfaces developed here (DOPA-P1@P2) can effectively modulate bone immunity, thereby creating a favorable immune microenvironment that supports subsequent stages of bone regeneration.

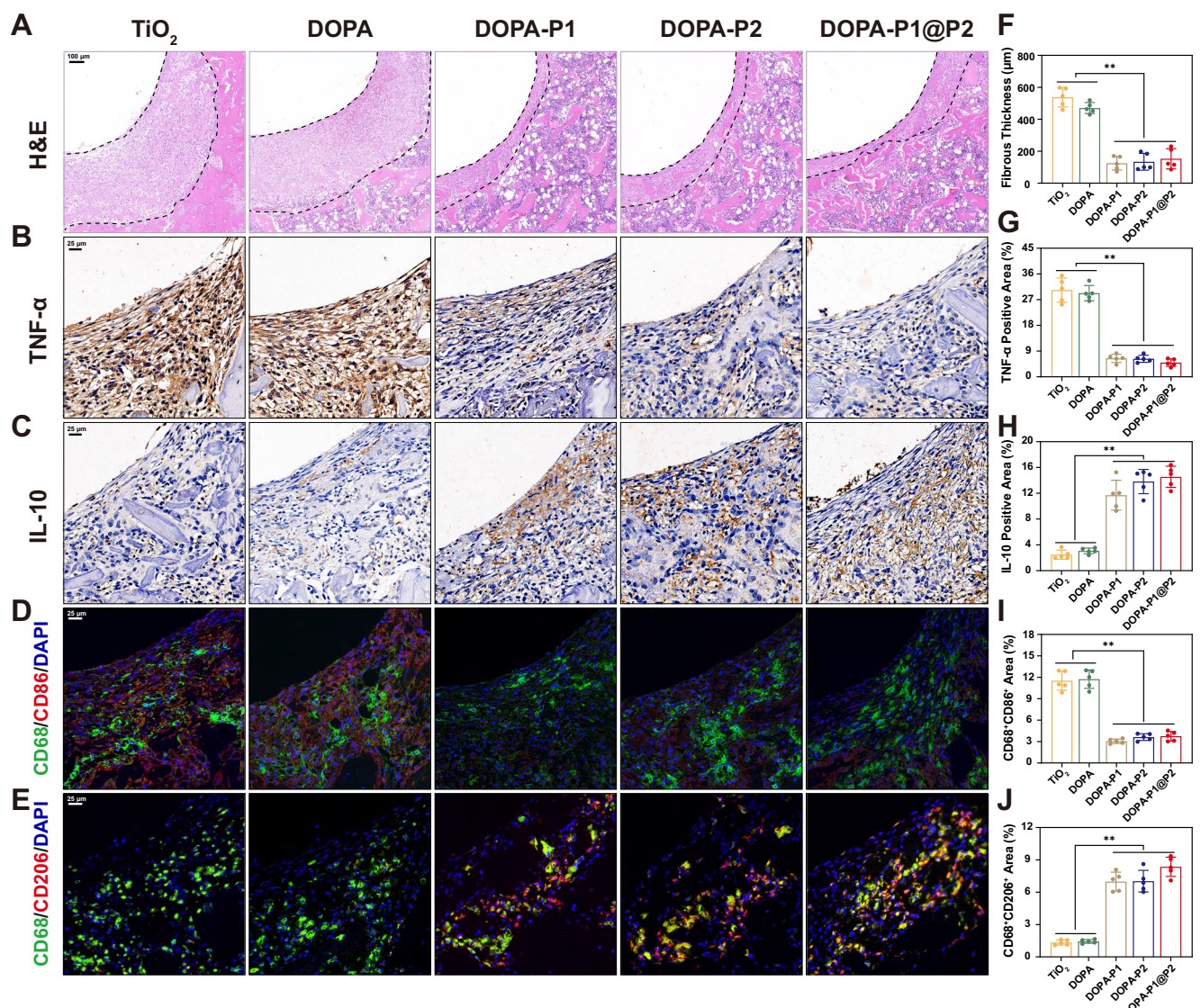

**Fig. 5 | Regulation of macrophage polarization in vivo. A** H&E staining of peri-implant tissue in the femur five days post-implantation, and **F** quantitative analysis of fibrous layer thickness (dashed line marks the peri-implant fibrous layer) ($n = 5$ per group). **B**, **C** Immunohistochemical staining of peri-implant tissue to assess TNF-α and IL-10 expression, with **G**, **H** quantitative results of the corresponding positive areas displayed ($n = 5$ per group). **D**, **E** Immunofluorescence staining of peri-implant tissue to evaluate macrophage polarization status, with green indicating the macrophage-specific marker (CD68), red representing markers for M1 (CD86) and M2 (CD206), and blue highlighting the nucleus. **I**, **J** Quantitative analysis results of immunofluorescence ($n = 5$ per group). The data are presented as the mean ± SD. Data were analyzed by one-way ANOVA with Tukey's post hoc test, and $*p < 0.05$ and $**p < 0.01$ indicate statistical significance. Exact $p$ values were given in the Source Data file.

## Indirect promotion of angiogenesis and osteogenesis

To verify that the developed modified surfaces can indirectly promote angiogenesis and osteogenesis through immunomodulatory effects, we employed conditioned medium (CM) from macrophages in subsequent experiments (Fig. 6A). Initially, to evaluate the migratory potential of HUVECs, we conducted a wound healing assay (Fig. 6B). Compared with those in the CM, TiO2$^{CM}$ and DOPA$^{CM}$ groups, the wound healing rates in the DOPA-P1$^{CM}$, DOPA-P2$^{CM}$, and DOPA-P1@P2$^{CM}$ groups, whose implant surfaces possess immunomodulatory properties, were significantly higher (Fig. 6I). Additionally, transwell assays further confirmed that following 24 h of incubation, increased cell migration was observed in the DOPA-P1$^{CM}$, DOPA-P2$^{CM}$, and DOPA-P1@P2$^{CM}$ groups compared to the CM, TiO2$^{CM}$ and DOPA$^{CM}$ groups (Fig. 6C, J). These results indicate that the CM extracted from surfaces modified with anti-inflammatory peptides effectively promoted the migration of HUVECs. Tube formation assays were conducted to assess

vascularization in the different groups at 4 and 24 h (Fig. 6D). Quantitative analysis revealed that, at both 4 and 24 h, the formation of branches and junctions was significantly increased in the DOPA-P1$^{CM}$, DOPA-P2$^{CM}$, and DOPA-P1@P2$^{CM}$ groups than in the CM, TiO2$^{CM}$ and DOPA$^{CM}$ groups (Fig. 6K, L), suggesting that surfaces modified with anti-inflammatory peptides can regulate the immune microenvironment to promote angiogenesis. The western blot results were consistent with these findings, showing significantly higher levels of angiogenesis-related proteins (VEGF and FGF-2) in groups with immunomodulatory implant surfaces (DOPA-P1$^{CM}$, DOPA-P2$^{CM}$, and DOPA-P1@P2$^{CM}$) compared to those without such properties (CM, TiO2$^{CM}$ and DOPA$^{CM}$) (Fig. 6E, M, N).

In addition to demonstrating that the developed modified surfaces can regulate immunity to enhance bone regeneration, confirming their capacity to indirectly promote osteogenesis is essential. Macrophage CM from different modified surfaces was combined with

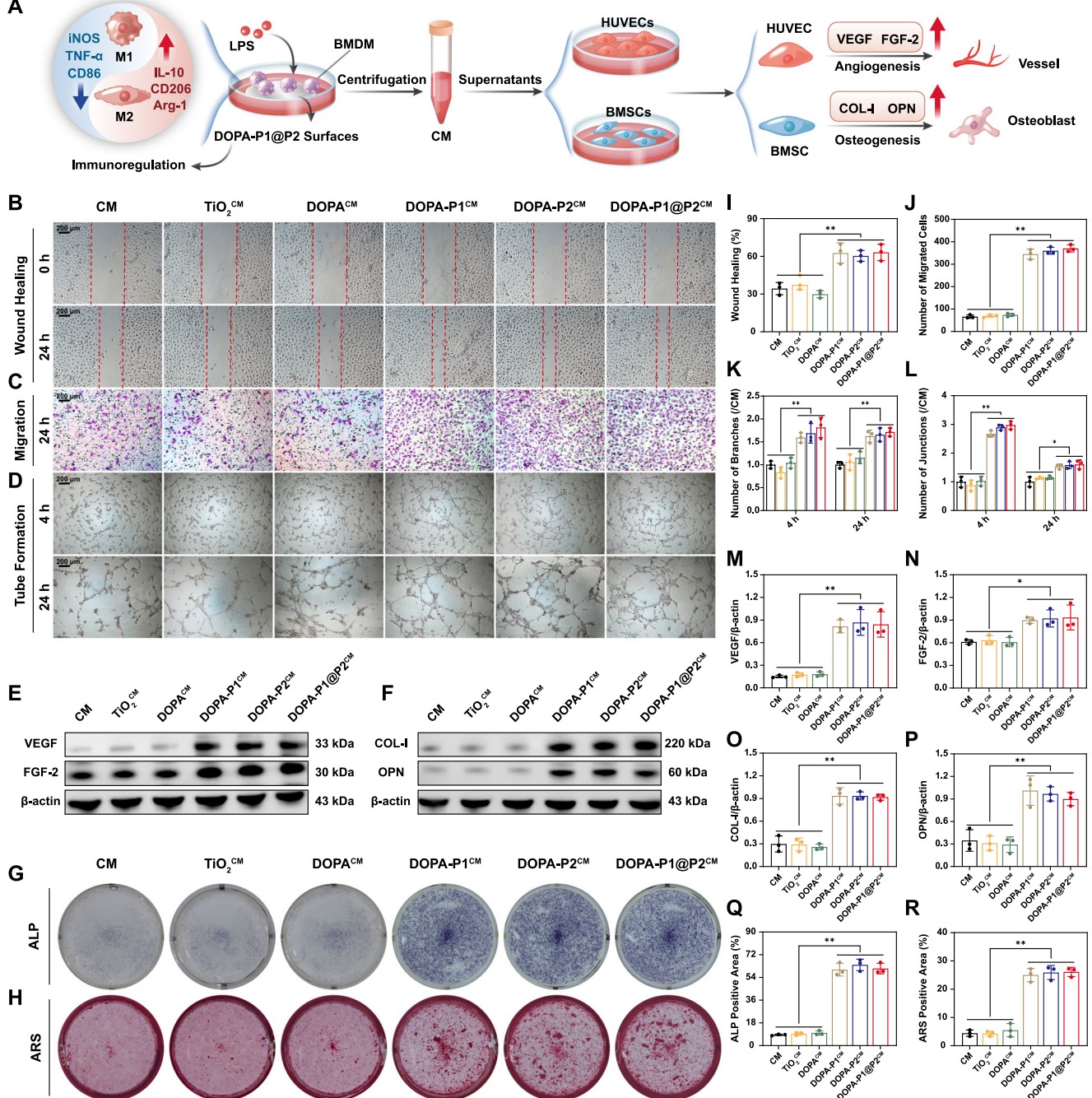

**Fig. 6 | Indirect promotion of angiogenesis and osteogenesis. A** The illustration of experimental design. **B** Wound healing assays were performed using conditioned medium, and **I** quantitative analysis of the results was conducted ($n = 3$ per group). **C** Transwell assays and corresponding **J** quantitative results ($n = 3$ per group). **D** Tube formation assays and corresponding **K, L** quantitative results ($n = 3$ per group). **E, F** WB analysis of angiogenic (VEGF and FGF-2) and osteogenic (COL-I and OPN) protein expression and **M–P** quantitative analysis results ($n = 3$ per group). **G, H** ALP and ARS staining and corresponding **Q, R** quantitative analysis results ($n = 3$ per group). The data are presented as the mean ± SD. Data were analyzed by one-way ANOVA with Tukey's post hoc test, and *$p < 0.05$ and **$p < 0.01$ indicate statistical significance. Exact $p$ values were given in the Source Data file.

osteogenic induction medium to culture BMSCs, with alkaline phosphatase (ALP) staining conducted on day 7 and alizarin red S (ARS) staining on day 21. ALP staining demonstrated that the area of positive staining for ALP was significantly increased in the DOPA-P1$^{CM}$, DOPA-P2$^{CM}$, and DOPA-P1@P2$^{CM}$ groups than in the CM, TiO$_2$$^{CM}$ and DOPA$^{CM}$ groups (Fig. 6G, Q). Similarly, ARS staining revealed that calcium nodule deposition was significantly increased in the groups with implants with immunomodulatory effects than in the other groups

(Fig. 6H, R). Western blot results verified that the levels of COL-I and OPN were markedly higher in the DOPA-P1$^{CM}$, DOPA-P2$^{CM}$, and DOPA-P1@P2$^{CM}$ groups compared to the CM, TiO$_2$$^{CM}$ and DOPA$^{CM}$ groups. (Fig. 6F, O–P). Additionally, the results of the immunofluorescence staining aligned with the aforementioned findings, indicating increased osteogenic activity in the modified surfaces with immunomodulatory effects (DOPA-P1$^{CM}$, DOPA-P2$^{CM}$, and DOPA-P1@P2$^{CM}$), as indicated by elevated OPN (depicted in red) expression

(Supplementary Fig. 6A–B). Finally, to validate the indirect angiogenic and osteogenic effects of the modified surfaces, assessments were also conducted at the gene level. RT-qPCR revealed that both angiogenesis-related genes (*VEGF* and *FGF2*) and osteogenesis-related genes (*Colla1* and *Opn*) were expressed at higher levels in the DOPA-P1$^{CM}$, DOPA-P2$^{CM}$, and DOPA-P1@P2$^{CM}$ groups than in the CM, TiO$_2$$^{CM}$ and DOPA$^{CM}$ groups (Supplementary Fig. 6C–F). In summary, the experimental results collectively indicated that surfaces modified with anti-inflammatory peptides effectively promote the polarization of macrophages towards the M2 phenotype, creating a favorable immunological microenvironment that indirectly enhances both angiogenesis and osteogenesis.

### Direct promotion of angiogenesis and osteogenesis

Although regulating the bone immune microenvironment can promote angiogenesis and osteogenesis to some extent, this indirect enhancement is inherently limited and difficult to sustain over long durations. To achieve satisfactory osseointegration at the implant interface, the surface must possess a direct and potent capacity to induce both angiogenesis and osteogenesis. In this work, we pretreated various modified surfaces with MMP-2, which specifically cleaves the PVGLIG peptide, to facilitate the release of anti-inflammatory peptides from the surface, thereby exposing the underlying sequences of angiogenic and osteogenic peptides (Fig. 7A). Initially, we conducted wound healing assays with HUVECs on modified titanium surfaces after pretreatment with MMP-2. The results demonstrated that surfaces modified with angiogenic peptides (DOPA-P1 and DOPA-P1@P2) significantly increased wound closure rates compared with those of the other groups (TiO$_2$, DOPA, and DOPA-P2) (Fig. 7B, I). We conducted HUVEC migration assays using different peptide solutions. Similarly, the results indicated that the groups with surfaces containing angiogenic peptides (DOPA-P1 and DOPA-P1@P2) had notably higher numbers of migrating cells compared to the other groups. (Fig. 7C, J). Tube formation assays were then conducted, and the findings demonstrated that at both the 4-h and 24-h time points, the DOPA-P1 and DOPA-P1@P2 groups exhibited a substantially higher number of branches and junctions compared to the other groups (Fig. 7D, K–L). Furthermore, HUVECs were cultured directly on titanium surfaces after pretreatment with MMP-2, followed by extraction of proteins and RNA for western blot and RT-qPCR analyses, respectively. Angiogenesis-related proteins (VEGF and FGF-2) and genes (*VEGF* and *FGF2*) were significantly upregulated in the DOPA-P1 and DOPA-P1@P2 groups compared with the TiO$_2$, DOPA, and DOPA-P2 groups (Fig. 7E, M–N and Supplementary Fig. 7E–F). These results demonstrate that titanium surfaces modified with angiogenic peptides (DOPA-P1 and DOPA-P1@P2) directly promote angiogenesis.

We subsequently investigated whether modified titanium surfaces could directly promote osteogenesis. BMSCs were directly cultured on modified titanium surfaces after pretreatment with MMP-2 for osteogenic induction, with ALP staining conducted on day 7 and ARS staining on day 21. The findings indicate that surfaces modified with osteogenic peptides (DOPA-P2 and DOPA-P1@P2) presented significantly higher ALP activity and calcium nodule deposition than the other surfaces did (TiO$_2$, DOPA, and DOPA-P1) (Fig. 7G, H, Q, R). Subsequent western blotting and immunofluorescence assays were performed to examine protein expression levels, revealing a significant elevation of COL-I and OPN in the DOPA-P2 and DOPA-P1@P2 groups compared to the other groups (Fig. 7F, O–P and Supplementary Fig. 7A, B). Finally, the results of gene expression analysis were consistent with the protein-level findings, with increased expression of *Colla1* and *Opn* in the DOPA-P2 and DOPA-P1@P2 groups relative to the TiO$_2$, DOPA, and DOPA-P1 groups (Supplementary Fig. 7C, D). These results suggest that surfaces modified with osteogenic peptides (DOPA-P2 and DOPA-P1@P2) possess a strong capacity to directly enhance osteogenesis. In summary, the inflammation-responsive

biomimetic surface (DOPA-P1@P2) developed in this study not only modulates the immune microenvironment to indirectly enhance angiogenesis and osteogenesis but also directly promotes these processes.

### Exploring mechanisms of angiogenesis and osteogenesis

To explore how the inflammation-responsive biomimetic modified surfaces directly enhance angiogenesis and osteogenesis, HUVECs and BMSCs were cultured on TiO$_2$ and DOPA-P1@P2 surfaces, and samples from both groups were subsequently collected for transcriptomic sequencing. The results of principal component analysis (PCA), box plots of gene expression levels, regional distribution plots, and heatmaps of correlation coefficients among samples, along with clustering analysis, indicated that the sequencing data for HUVECs and BMSCs met the required standards, confirming the reliability of the results (Fig. 8A, G and Supplementary Fig. 8A–H). Volcano plot analysis revealed that, in HUVECs, compared with the TiO$_2$ group, the DOPA-P1@P2 group presented 411 upregulated genes and 557 downregulated genes (Fig. 8B). In BMSCs, compared with the TiO$_2$ group, the DOPA-P1@P2 group presented 851 upregulated and 976 downregulated genes (Fig. 8H). As depicted in the heatmaps in Fig. 8C, I, notable differences were observed in the expression patterns of differentially expressed genes (DEGs) between the TiO$_2$ and DOPA-P1@P2 groups. The radar chart of the differentially expressed genes further confirmed the above results (Supplementary Fig. 9A, D). These results indicate that the inflammation-responsive biomimetic modified surface developed here (DOPA-P1@P2) substantially affects the proliferation and differentiation processes of HUVECs and BMSCs.

After sequencing of the HUVEC samples, Gene Ontology (GO) enrichment analysis suggested that a considerable portion of the DEGs were involved in the regulation of angiogenesis (Fig. 8E and Supplementary Fig. 9B). Moreover, findings from Kyoto Encyclopedia of Genes and Genomes (KEGG) enrichment analysis showed that the Mitogen-Activated Protein Kinase (MAPK) signaling pathway may play a pivotal role in modulating angiogenesis (Fig. 8F and Supplementary Fig. 9C). Furthermore, the results from gene set enrichment analysis (GSEA) showed that the MAPK signaling pathway was significantly upregulated in the DOPA-P1@P2 group in comparison to the TiO$_2$ group (Fig. 8D). However, in the GO enrichment analysis of the BMSC samples, we found that numerous DEGs were enriched in processes such as the extracellular matrix and bone mineralization, which are crucial for osteogenesis (Fig. 8K and Supplementary Fig. 9E). Furthermore, the results from the KEGG enrichment analysis revealed that the differentially expressed gene sets were enriched in the Phosphoinositide 3-Kinase–Akt (PI3K–Akt) signaling pathway, with GSEA further indicating significant activation of this pathway in the DOPA-P1@P2 group in comparison to the TiO$_2$ group (Fig. 8J, L and Supplementary Fig. 9F). Consequently, it is possible that DOPA-P1@P2 surfaces promote angiogenesis and osteogenesis through activation of the MAPK and PI3K–Akt signaling pathways, respectively.

To further investigate the underlying mechanisms, we performed western blot analysis. The results suggest that the pro-angiogenic effects of DOPA-P1@P2 surfaces may be associated with the activation of extracellular signal-regulated kinase 1/2 (ERK1/2) within the MAPK signaling pathway, while their pro-osteogenic effects are likely linked to the activation of mechanistic target of rapamycin (mTOR) downstream of the PI3K–Akt signaling pathway (Supplementary Fig. 10A–D).

### Evaluation of interfacial osseointegration

Our in vitro investigations revealed that DOPA-P1@P2 surfaces are capable of modulating immune responses while facilitating angiogenesis and osteogenesis. However, whether the DOPA-P1@P2 surface can similarly exert these effects in vivo, thereby achieving stable osseointegration at the interface, is a key focus of this study. This question highlights the primary innovation of our research, which is

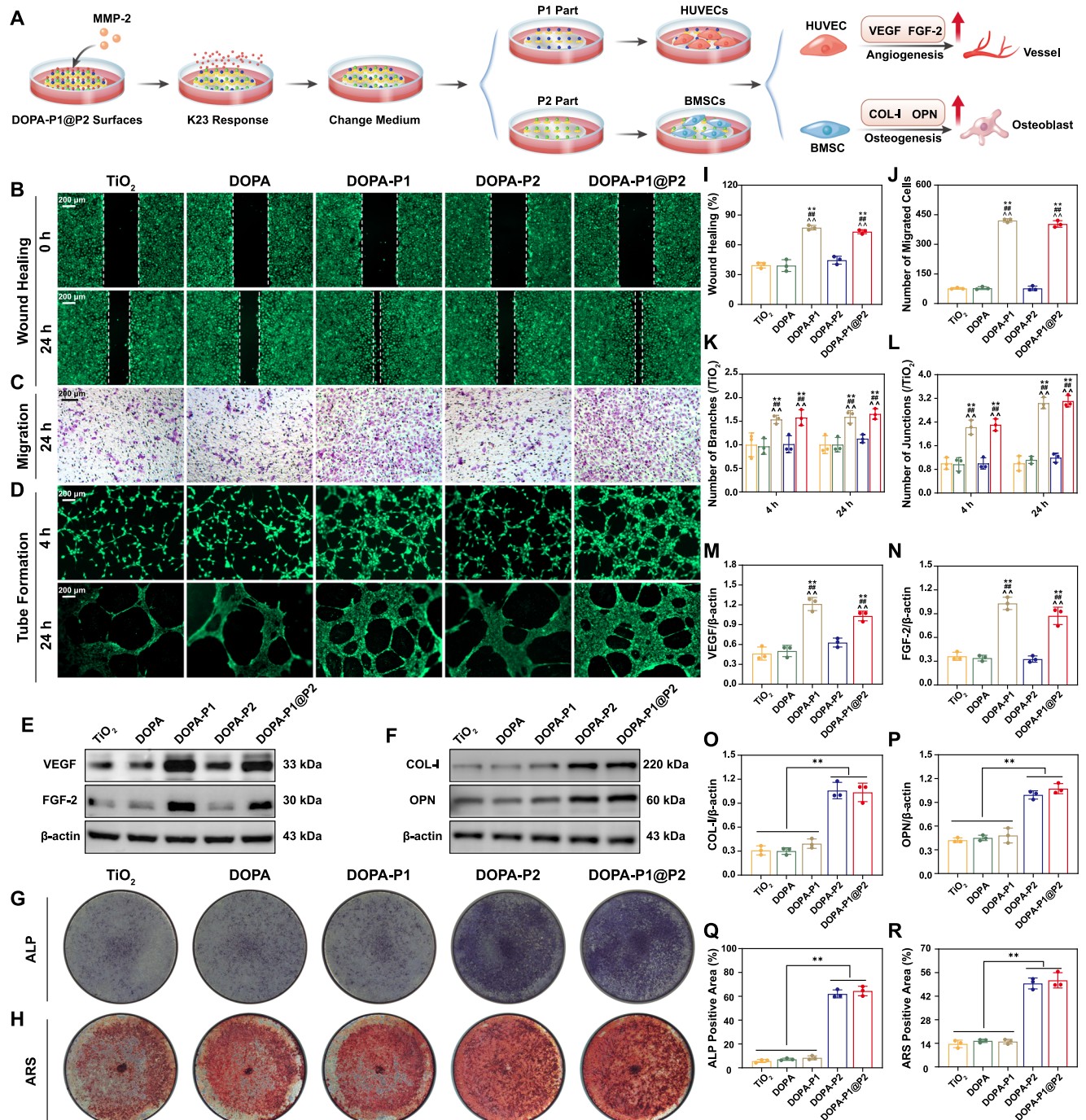

**Fig. 7 | Direct promotion of angiogenesis and osteogenesis. A** The illustration of experimental design. **B** Wound healing assays were conducted directly on the modified titanium surfaces, along with the corresponding **I** quantitative analysis results (n = 3 per group). **C** Transwell assays and corresponding **J** quantitative results (n = 3 per group). **D** Tube formation assays and corresponding **K**, **L** quantitative results (n = 3 per group). **E**, **F** WB analysis of angiogenic (VEGF and FGF2) and osteogenic (COL-I and OPN) protein expression and **M–P** quantitative analysis results (n = 3 per group). **G**, **H** ALP and ARS staining of modified titanium surfaces, along with the corresponding **Q**, **R** quantitative analysis results (n = 3 per group). The data are presented as the mean ± SD. Data were analyzed by one-way ANOVA with Tukey's post hoc test (*$p < 0.05$ and **$p < 0.01$ vs. the TiO₂ group or comparisons between combined groups indicated in the figure; #$p < 0.05$ and ##$p < 0.01$ vs. the DOPA group; ^$p < 0.05$ and ^^$p < 0.01$ vs. the DOPA-P2 group$p$). $p < 0.05$ and $p < 0.01$ indicate statistical significance. Exact $p$ values were given in the Source Data file.

the development of inflammation-responsive modified titanium surfaces that regulate the process of bone regeneration in an orderly manner to achieve satisfactory osseointegration. Here, we implanted titanium rods with various modifications into the femurs of rats and collected the femurs for micro-CT and histological examinations after 8 weeks (Fig. 9A). Three-dimensional images reconstructed from micro-CT scans clearly demonstrated that the bone mass around the titanium rods was highest in the DOPA-P1@P2 group, followed by the DOPA-P2 and DOPA-P1 groups (Fig. 9B). The quantitative bone mineral density (BMD) results clearly showed that the DOPA-P1@P2 group presented the highest BMD (1.31 ± 0.09 g/cm³), outperforming the DOPA-P2 (0.96 ± 0.06 g/cm³) and DOPA-P1 (0.86 ± 0.05 g/cm³) groups, whereas very low BMD were recorded in the DOPA (0.55 ± 0.04 g/cm³) and TiO₂ (0.55 ± 0.03 g/cm³) groups (Fig. 9E).

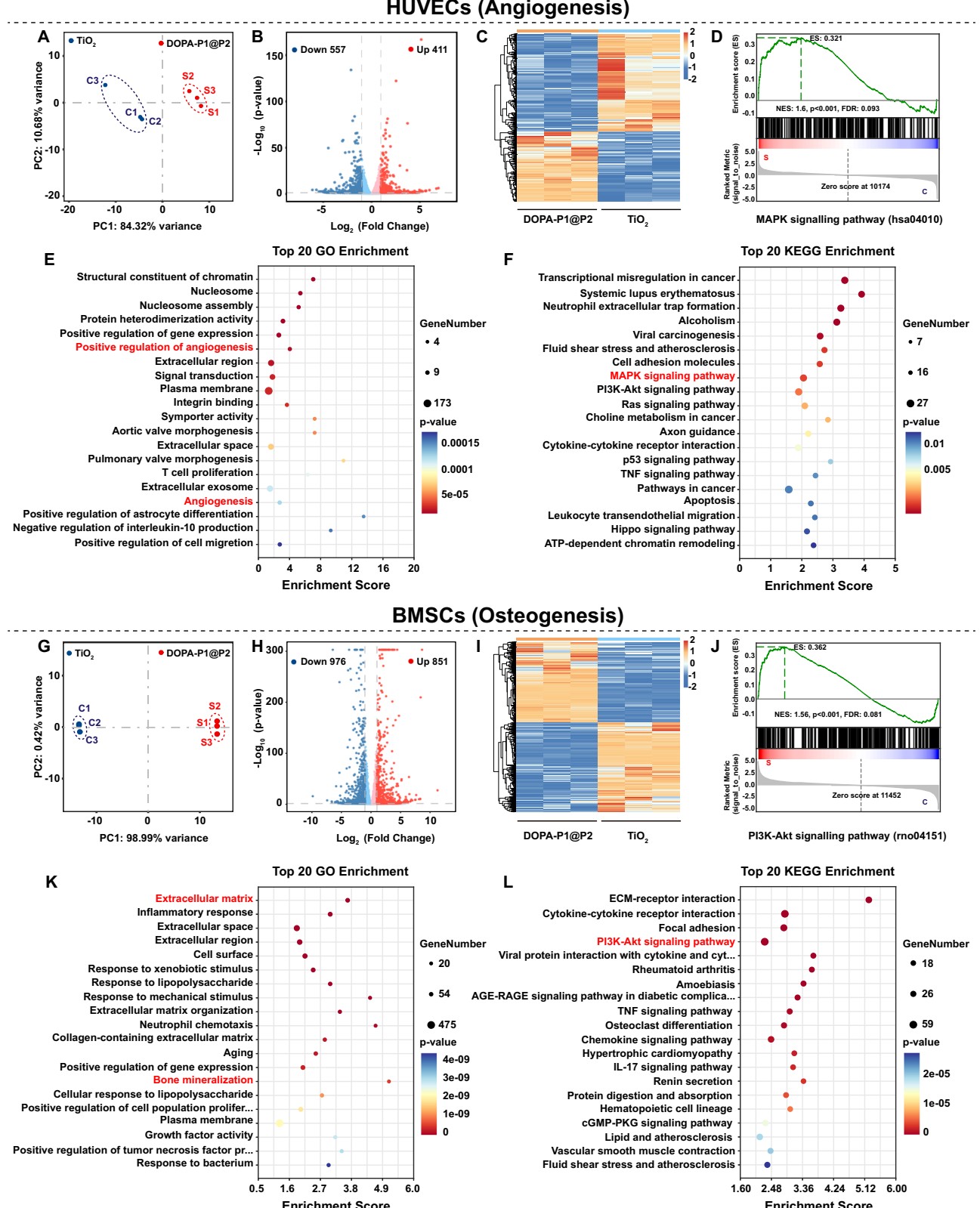

**Fig. 8 | Transcriptome sequencing for exploring the mechanisms of angiogenesis and osteogenesis.** **A**, **G** Principal component analysis (PCA) of differentially expressed genes (DEGs) in the TiO₂ and DOPA-P1@P2 groups. **B**, **H** Volcano plots of DEGs. **C**, **I** Heatmap of DEGs. **E**, **K** TOP 20 Gene Ontology (GO) enrichment analysis of DEGs. **F**, **L** TOP 20 Kyoto Encyclopedia of Genes and Genomes (KEGG) enrichment analysis of DEGs. **D**, **J** Gene set enrichment analysis (GSEA) of the MAPK and PI3K–Akt signaling pathways. The data are presented as the mean ± SD. Statistical analysis was performed by unpaired two-tailed Student's *t*-test.

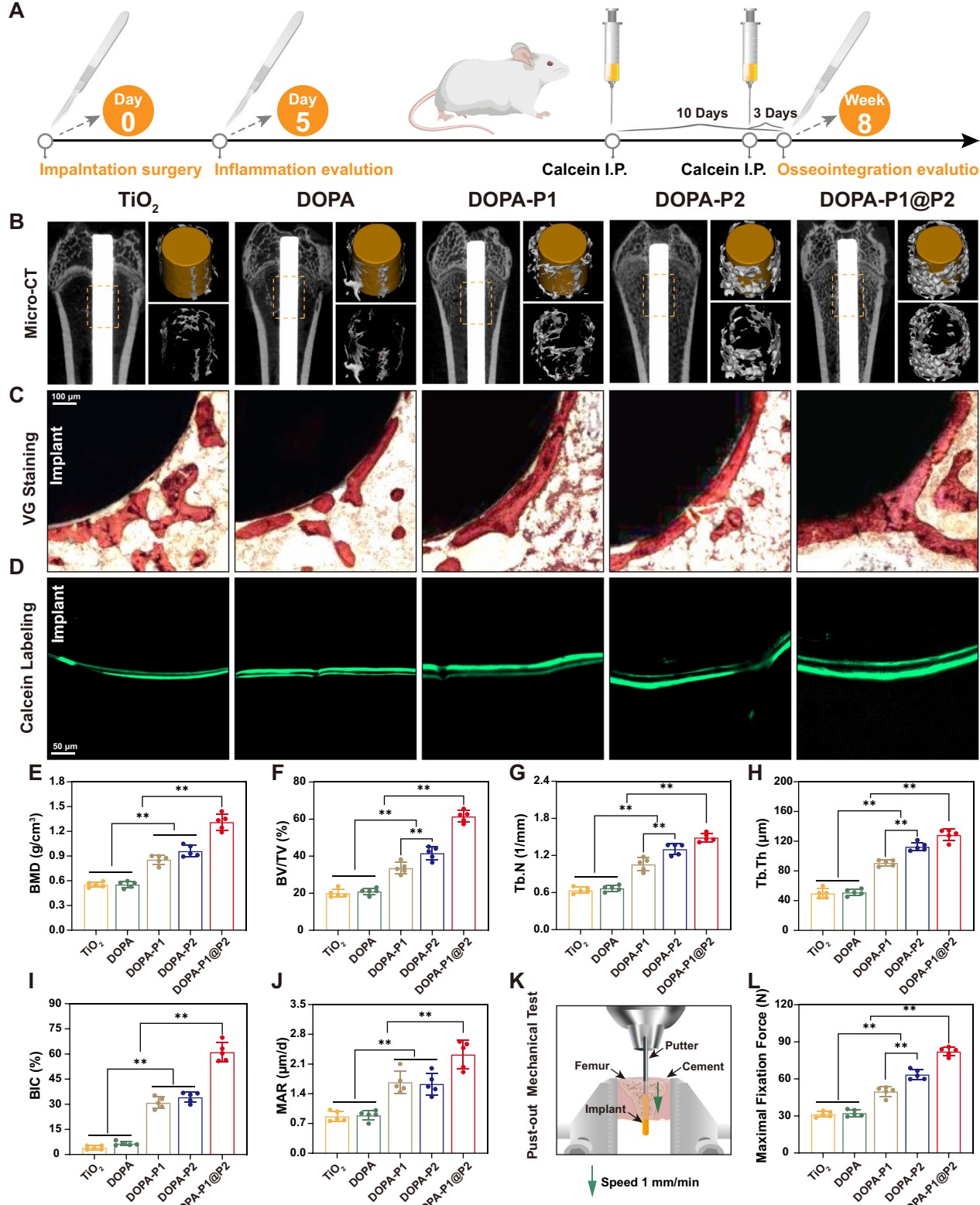

**Fig. 9 | DOPA-P1@P2 effectively enhances interfacial osseointegration in vivo.**
**A** The illustration of experimental design. **B** Micro-CT scans and three-dimensional reconstruction images, accompanied by the corresponding **E–H** quantitative analysis results ($n = 5$ per group). **C** Van Gieson's staining of hard tissue sections containing implants, accompanied by the **I** quantitative analysis results of the bone-implant contact ratio (BIC) ($n = 5$ per group). **D** Calcein double-labeling experiment of newly formed bone around the implant, along with **J** quantitative analysis results

of bone mineralization deposition rate (MAR) ($n = 5$ per group). **K** Schematic diagram illustrating the biomechanical push-out experiment, accompanied by corresponding **L** quantitative analysis results ($n = 5$ per group). The data are presented as the mean ± SD. Data were analyzed by one-way ANOVA with Tukey's post hoc test, and *$p < 0.05$ and **$p < 0.01$ indicates statistical significance. Exact $p$ values were given in the Source Data file.

Additionally, the quantitative results for the trabecular bone parameters (BV/TV, Tb.N, and Tb.Th) further confirmed the improved osseointegration in the DOPA-P1@P2 group (Fig. 9F–H). For example, the bone volume fraction (BV/TV) in the DOPA-P1@P2 group reached $61.44 \pm 2.83\%$, which was 1.47-fold higher than that of the DOPA-P2 group, 1.83-fold higher than that of the DOPA-P1 group, and over 3-fold higher than that of the $TiO_2$ group.

Following micro-CT analysis, we conducted undecalcified bone slicing and van Gieson (VG) staining of femurs containing titanium rods to assess the bone-implant contact ratio (BIC) across various groups. As expected, the DOPA-P1@P2 group presented a bone-to-implant contact (BIC) value of $61.10 \pm 5.12\%$, which was markedly higher than the measurements observed for all other groups, with the BIC values in the DOPA-P1 and DOPA-P2 groups also showing marked enhancements over those in the $TiO_2$ and DOPA groups (Fig. 9C, I). Additionally, dual labeling with calcein confirmed that the mineral apposition rate (MAR) of the DOPA-P1@P2 group was markedly higher compared with the other groups, demonstrating its potential to promote osteogenesis (Fig. 9D, J). The mechanical push-out test (Fig. 9K) is a critical method for evaluating the integration of an implant with surrounding bone tissue[27]. We observed that DOPA-P1@P2 achieved the highest mechanical push-out force ($82.12 \pm 3.12$ N), which was substantially higher than the values observed in the other groups (Fig. 9L). This force was 1.29 times higher than that of the DOPA-P2 ($63.52 \pm 3.58$ N) group, 1.65 times higher than that of the DOPA-P1 ($49.76 \pm 3.67$ N) group, and more than 2.5 times higher than that observed in the DOPA ($32.31 \pm 2.41$ N) and $TiO_2$ ($31.35 \pm 2.05$ N) groups, suggesting that the osseointegration achieved in the DOPA-P1@P2 group was the strongest among the tested groups. Notably, the mechanical push-out strength of the DOPA-P1@P2 group was notably higher than that of the DOPA-P1 and DOPA-P2 groups, likely due to the synergistic effects of angiogenesis and osteogenesis achieved on the DOPA-P1@P2 surface. Finally, a subset of femurs containing titanium rods was decalcified, after which the titanium rods were carefully removed for subsequent Masson and immunohistochemical staining. In alignment with the aforementioned findings, the DOPA-P1@P2 group demonstrated the highest collagen volume fraction and the highest levels of COL-I and OPN in comparison to the other groups (Supplementary Fig. 11A–F). In summary, the DOPA-P1@P2 surface can regulate the process of bone regeneration in an orderly manner, enhancing the synergistic effects of angiogenesis and osteogenesis and thereby achieving satisfactory osseointegration at the interface.

## Discussion

Aseptic loosening due to insufficient osseointegration at the bone–implant interface remains a leading cause of prosthetic failure after joint arthroplasty[28]. The surface modification of bone implants to enhance bioactivity has shown promise in improving interfacial osseointegration[29]. Early efforts focused on enhancing the osteoinductive properties of implant surfaces[30], but targeting osteogenesis alone was inadequate. Successful implant-mediated bone regeneration requires the coordinated interplay of immunomodulation, angiogenesis, and osteogenesis, which are temporally regulated. While emerging implant surfaces can modulate these processes simultaneously[31], insufficient attention to their temporal dynamics leads to disordered interactions, requiring further optimization. In this study, we developed an inflammation-responsive biomimetic titanium surface (DOPA-P1@P2) that effectively promotes bone regeneration by intentionally aligning with the physiological process. Previous studies reported that[32,33], compared with $TiO_2$ control, the maximal push-out force increased by an average of 150%, bone volume to total volume (BV/TV) increased by 120%, and bone-to-implant contact (BIC) increased by 300%. In this work, DOPA-P1@P2 achieved a 161% increase in the maximal push-out force, a 207% increase in BV/TV, and an extraordinary 1409% increase in BIC. These findings highlight notable advancements attributed to the design concept that aligns with the physiological processes of bone regeneration to enhance osseointegration.

Traditionally, surface modification of biomaterials relies on incorporating bioactive macromolecules, such as VEGF and BMP-2, or metal ions, such as $Zn^{2+}$ and $Mg^{2+}$, to achieve functionalization[34–37]. However, these macromolecules are difficult to chemically modify and prone to inactivation, and the toxicity and uncontrolled release of metal ions limit their application[38,39]. Recent advances in peptide synthesis have positioned peptides as promising alternatives for biomaterial surface modification, offering comparable bioactivity, increased safety, increased stability, and enhanced feasibility of chemical modification[40]. Thus, in this work, we introduced four functional peptides to modify titanium-based surfaces. K23 (KAFAKLAARLYR-KALARQLGVAA) is an anti-inflammatory, cell-penetrating peptide (CPP) that inhibits proinflammatory cytokines such as IL-1, IL-6, and TNF-α by targeting mitogen-activated protein kinase–activated protein kinase 2 (MK2)[41]. It has been shown to reduce inflammation in a macrophage model with human monocytes[42], making it a suitable choice for its anti-inflammatory role in this study. K15 (KLTWQE-LYQLKYKGI), a synthetic 15-amino-acid polypeptide derived from the VEGF binding region with VEGF-like angiogenic activity, activates endothelial cells to mimic VEGF functions such as invasion, chemotaxis, and capillary sprouting[43], making it the ideal choice for promoting angiogenesis in this work. Y5 (YGFGG), the C-terminal fragment of osteogenic growth peptide (OGP), is the active component that enhances osteoblast proliferation and differentiation, was utilized in this study to promote osteogenesis[26]. Macrophages within the inflammatory microenvironment regulate pathophysiological processes through the secretion of matrix metalloproteinases, particularly MMP-2 and MMP-9[44]. The peptide sequence PVGLIG, which is specifically cleaved by either MMP-2 or MMP-9, serves as an ideal linker between K23 and K15, as well as between K23 and Y5, enabling an inflammation-responsive effect during the first stage of bone regeneration[24].

To date, physical and chemical modification techniques have been widely employed in the field of biomaterial surface engineering. For example, methods such as layer-by-layer self-assembly, anodization, acid etching, and ion doping are commonly utilized to introduce bioactive molecules onto implant surfaces[45–48]. However, these physical approaches often suffer from significant molecular leakage and face challenges in maintaining long-term bioactivity[30], thereby limiting their practical application. On the other hand, conventional chemical modification methods often involve complex reactions and tedious surface treatment procedures, with limited controllability and operability, particularly in the development of multifunctional surface modifications[49]. Thus, developing a simple and efficient surface modification technique capable of conjugating multiple bioactive molecules is critically important. Inspired by the adhesion mechanism of marine mussel foot proteins (Mfps), mussel-inspired bioinspired strategies have emerged as highly promising surface modification approaches, leveraging the repetitive catecholamine structure of DOPA in Mfps to achieve strong adhesion on nearly all types of surfaces[32]. Moreover, DOPA can effectively immobilize various bioactive molecules through covalent bonding while significantly enhancing the biocompatibility of material surfaces[7,30]. Given these advantages, mussel-inspired biomimetic strategies represent a approach for surface modification in this study. However, when mussel-inspired peptides are used to modify titanium surfaces, the coordination between DOPA and $TiO_2$ may compromise the activity of biomolecules covalently linked to DOPA[33]. Consequently, there is an urgent need to optimize the existing mussel-inspired surface modification strategies. Fortunately, the rise of bioorthogonal click chemistry has opened a avenue, as its mild and direct reaction conditions, rapid reaction rates, and specificity make this technique a current research hotspot in the

field of implant surface modification[50]. The surface modification approach employed in this work combines mussel-inspired biomimetic strategies with bioorthogonal click chemistry, overcoming numerous drawbacks of traditional surface modification techniques. This method is straightforward, practical, and economical and holds promise for widespread application in settings requiring multifunctional surface modifications.

The primary prerequisite for the clinical application of implanted materials is their nontoxicity to the body[51]. This study demonstrated the cytocompatibility of DOPA-P1@P2 in three cell types (BMDMs, HUVECs, and BMSCs) and confirmed its in vivo safety for rat visceral organs, establishing a foundation for its practical application. Previous research has demonstrated that increased surface roughness enhances osteoblast adhesion and spreading, which facilitates osseointegration compared with smooth surfaces[52]. Furthermore, enhancing the hydrophilicity of material surfaces promotes the maintenance of protein structure and function, enhances cellular adhesion, and improves biocompatibility[53]. Compared with those of the $TiO_2$ control, the surface roughness and hydrophilicity of the biomimetic surfaces (DOPA-P1@P2) constructed in this work were significantly improved, facilitating osteoblast adhesion and spreading, which is critical for improving interfacial bone regeneration.

The initial stage of bone regeneration mediated by implant materials in vivo is characterized predominantly by an inflammatory response[54]. During this phase, a substantial hematoma forms on the implant surface, dominated by M1 macrophages, which release a plethora of inflammatory mediators, serving as a protective response against injury[55]. However, the resolution of acute inflammation during the early postimplantation period, particularly the transition from the proinflammatory M1 phenotype to the anti-inflammatory M2 phenotype before the proliferation stage, is a critical determinant of the fate of implant materials in the body[56]. The prolonged dominance of M1 macrophages at the implant site leads to the sustained release of inflammatory mediators, which results in a shift from acute inflammation postimplantation to chronic inflammation. This triggers the foreign body response, with the recruitment of fibroblasts to encapsulate the implant material with a fibrous capsule, which impairs osseointegration and potentially leads to implant failure[57]. Conversely, the transition from the M1 to the M2 macrophage phenotype before the proliferation stage can steer bone regeneration in the right direction. By secreting essential chemokines and growth factors, M2 macrophages enhance angiogenesis and the migration, homing, and osteogenic differentiation of BMSCs, thus optimizing conditions for subsequent stages of bone regeneration[58]. According to the literature, the inflammatory response peaks within the first 24 h of the inflammation stage, lasting approximately 7 days before transitioning to the proliferation stage[59]. We selected the fifth day postimplantation (late inflammation stage) to assess whether the material effectively promoted M2 polarization prior to the proliferation phase, which is consistent with previous reports[32,33]. In vivo and in vitro experiments revealed that DOPA-P1@P2 effectively modulates the immune microenvironment and promotes macrophage M2 polarization, achieving 79.50% inhibition of M1 polarization, as shown by flow cytometry analysis, and demonstrated enhanced performance compared with related studies[7,33].

Angiogenesis and osteogenesis jointly constitute the proliferation stage of bone regeneration, which lasts 3 to 4 weeks[12]. Neovascularization provides a vital connection between bone and the surrounding tissues, enabling the transport of oxygen, nutrients, and BMSCs to the injured area[60]. The activation of the VEGF signaling pathway has been reported in the literature to effectively promote angiogenesis[61]. The angiogenic peptide K15 used in this study is a VEGF-mimetic peptide that effectively promotes angiogenesis and activates the MAPK signaling pathway, which is consistent with previous reports[62,63]. BMSCs around the injury site differentiate into osteoblasts in the local microenvironment, with osteogenic growth peptide (OGP) significantly increasing their proliferation and differentiation[64]. Consistent with previous reports[65,66], the osteogenic peptide Y5 used in this study effectively promoted osteogenesis and activated the PI3K–Akt signaling pathway. However, angiogenesis and osteogenesis, both critical processes in bone regeneration, must be effectively integrated to achieve optimal interface osseointegration. In this study, the DOPA-P1@P2 group achieved enhanced osseointegration outcomes compared with the groups promoting angiogenesis (DOPA-P1) or osteogenesis (DOPA-P2) alone, likely due to the synergistic effects of angiogenesis and osteogenesis.

This study has several limitations, such as the inability of our experimental conditions to fully replicate the complexities of the physiological environment, including dynamic fluid flow, pH variations, and the influence of biological components (e.g., enzymes, proteins, and cells) on material behavior. Additionally, the coupling mechanisms between angiogenesis and osteogenesis have not been explored further. In future studies, we will incorporate advanced materials (e.g., thermoresponsive or pH-responsive materials) to increase the responsiveness. and precision of the coatings. Additionally, we will investigate the interaction mechanisms between HUVECs and BMSCs to gain deeper insights into the physiological processes of bone regeneration.

In summary, we developed an inflammation-responsive biomimetic titanium surface (DOPA-P1@P2) using a mussel-inspired strategy combined with bioorthogonal click chemistry. Upon implantation, the outermost K23 layer modulates acute inflammation during the first stage of bone regeneration, facilitating macrophage polarization from the M1 phenotype to the M2 phenotype. Activated macrophages continuously secrete MMP-2 and MMP-9, which specifically cleave PVGLIG sequences. This leads to detachment of the outermost K23 layer, exposing the underlying K15 and Y5 layers. These layers promote angiogenesis and osteogenesis during the second stage, followed by bone remodeling. This inflammation-responsive biomimetic surface not only supports immunomodulation, angiogenesis, and osteogenesis during bone regeneration but also coordinates these functions across different stages, achieving improved osseointegration at the bone–implant interface. This design concept for multifunctional surface modification holds promise for clinical translation and application.

## Methods

### Ethical statement

All procedures involving animals strictly adhered to animal welfare guidelines and were approved by the Ethics Committee of Union Hospital, Tongji Medical College, Huazhong University of Science and Technology, with the corresponding approval number: [2022] IACUC number: 3625.

### Animal models

Eighty 8-week-old SD rats were evenly divided into five groups: the $TiO_2$, DOPA, DOPA-P1, DOPA-P2, and DOPA-P1@P2 groups. The rats were anaesthetized via isoflurane inhalation, followed by the implantation of differently modified titanium rods into both femurs of the rats. The detailed surgical procedure is shown in Supplementary Fig. 12A–F. Five days postsurgery, six rats per group were euthanized, and their femurs were collected for inflammation assessment. The remaining ten rats in each group were administered an intraperitoneal injection of 10 mg/kg calcein (Sigma) 10 days and 3 days prior to euthanasia. All the rats were euthanized two months after surgery, and their bilateral femurs, hearts, livers, spleens, lungs, and kidneys were obtained for additional research.

## Peptide synthesis and surface modification

The mussel-inspired peptide $(DOPA)_4$-$OEG_5$-DBCO, which features multiple DOPA units and DBCO groups, and two azide-modified composite functional peptides, P1 ($N_3$-K15-PVGLIG-K23) and P2 ($N_3$-Y5-PVGLIG-K23), were produced using the Fmoc solid-phase synthesis approach by ChinaPeptides Co., Ltd. (QYAOBIO). The corresponding molecular designs are as follows: Ac-(DOPA)-G-(DOPA)-K[($OEG_5$)-(DBCO-COOH)]-(DOPA)-G-(DOPA), (2-Azido)-KLTWQELYQLKYKGIP VGLIGKAFAKLAARLYRKALARQLGVAA, and (2-Azido)-YGFGGPVGLIG KAFAKLAARLYRKALARQLGVAA. Here, OEG was used as a spacer to reduce steric hindrance and facilitate reactions, whereas DBCO and Azido functioned as the groups that facilitated bioorthogonal reactions. Titanium plates, 1 mm in thickness and 15 mm in diameter, along with titanium rods, 10 mm long and 1.5 mm in diameter, were procured from Tianjin Zhengtian Medical Devices Co., Ltd., China. After being treated with oxygen plasma, the titanium rods and plates were cleaned three times with acetone, 75% alcohol, and deionized water, followed by sterilization under high temperature and pressure. To minimize amino acid oxidation, all peptide solutions were purged with nitrogen for 15 min prior to use. Initially, sterile titanium plates and rods were submerged in a 0.01 mg/mL solution of $(DOPA)_4$-$OEG_5$-DBCO for 48 h to coat the titanium surface with mussel-inspired peptides through the spontaneous coordination of catechol groups and metal ions. The DBCO group in mussel-inspired peptides provides a click site for the subsequent grafting of peptides. Subsequently, surfaces modified with mussel-inspired peptides (defined as the DOPA group) were separately immersed in 0.1 mg/mL solutions of P1, P2, or a 1:1 mixture of both for 4 h, forming different modified surfaces (defined as the DOPA-P1, DOPA-P2, and DOPA-P1@P2 groups, respectively) through bioorthogonal click chemistry reactions between -DBCO and -$N_3$. The plates or rods immersed in PBS were designated the $TiO_2$ group and served as the control. The surfaces with different modifications were meticulously cleaned with MiniQ water and subsequently dried under nitrogen.

## Material characterization

The purities of $(DOPA)_4$-$OEG_5$-DBCO, P1 ($N_3$-K15-PVGLIG-K23) and P2 ($N_3$-Y5-PVGLIG-K23) were evaluated via HPLC, and their corresponding molecular weights were measured via ESI−MS. The presence of specific catechol groups in the mussel-inspired peptides was subsequently detected via $^1$H NMR. AFM was utilized to evaluate the surface morphology and quantify the roughness of the different groups. The hydrophilicity of the different groups was evaluated by measuring the static water contact angle. XPS was used to determine the elemental compositions of the different modified surfaces. The P1 and P2 surfaces were immersed in high-glucose DMEM without MMP-2 and with 1.5 µg/mL MMP-2, respectively, and incubated at 37 °C. The supernatants were collected at various time points, and the release trends of the PVGLIG peptide sequences were tracked using HPLC.

## Cell culture

In this work, three distinct cell types were utilized for experimental investigation. HUVECs were supplied by the Institute of Biochemistry and Cell Biology of the Chinese Academy of Sciences (Shanghai, China). BMSCs were isolated from the femoral marrow of 4-week-old SD rats. In brief, the extraction of BMSCs required a relatively sterile environment. The bilateral femurs of the rats were collected, and the adjacent soft tissues were carefully removed. Subsequently, scissors were used to carefully cut both ends of the femur, and a 1 ml syringe was used to flush the marrow cavity with culture medium repeatedly until the flushed medium became clear. The mixture was subsequently gathered and passed through a mesh filter. After filtration, the cell suspension was exposed to red blood cell lysis solution and then centrifuged. The resulting cells were resuspended and cultured in a dish. When the cell density in the culture dish reached approximately 80%, the cells were

digested and passaged with 0.25% trypsin/EDTA solution (RG-CE-9, Ketu Biotech). BMDMs were isolated from the bone marrow of 6-week-old C57BL/6 mice following a procedure similar to that used for BMSCs. M-CSF (20 ng/mL) was added to the culture medium to induce macrophage differentiation following cell seeding. HUVECs were cultured in high-glucose DMEM supplemented with 10% fetal bovine serum (FBS), with cell passages typically occurring every 1–2 days. BMSCs were cultured in low-glucose DMEM supplemented with 10% FBS, with cell passage typically performed every 2–3 days. The BMDMs were cultured in high-glucose complete DMEM, with half the medium replaced every 2–3 days, and fresh M-CSF was added. After 7 days, the cells were differentiated into macrophages for subsequent experiments.

## Cytocompatibility

Three types of cells were seeded onto differently modified titanium surfaces and cultured for 24 h, followed by staining using the Calcein−AM/PI Double Stain Kit (Yeasen, China). The cells were cultured on different modified surfaces for 24 h, and cytotoxicity was assessed via an LDH Cytotoxicity Assay Kit (Beyotime, China). Furthermore, the cells were cultured on diverse titanium surfaces, and cell proliferation was evaluated at 24 and 72 h via a Cell Counting Kit-8 (Yeasen, China). Finally, we performed cytoskeletal staining with phalloidin (Yeasen, China) to assess the adhesion and spreading characteristics of BMSCs on different modified surfaces. Briefly, BMSCs cultured on various modified surfaces for 12 h were fixed with 4% paraformaldehyde for 20 min, incubated with FITC-labeled phalloidin for 30 min, stained with DAPI for 10 min, and subsequently imaged using a confocal microscope (Zeiss, LSM800, Germany). Semiquantitative analysis was performed using ImageJ software (version 1.54 d), with all the data analysed in triplicate.

## Macrophage polarization

BMDMs were distributed across various surfaces at 20,000 cells per surface and subjected to 8 h of treatment with 100 ng/mL lipopolysaccharide (LPS) sourced from Sigma-Aldrich. Fresh DMEM was then added after replacing the old medium, and the cells were left to incubate for a further 48-h period. Following a 48-h incubation period, the supernatants from each group were collected, and ELISA kits were used to quantify the levels of TNF-α, IL-10, VEGF, and BMP-2. Simultaneously, the collected cells were subjected to immunofluorescence staining to assess their polarization characteristics on different modified surfaces. The immunofluorescence staining procedure followed the instructions of the TSA kit (ABclonal, RK05903). Primary antibodies against F4/80 (Abcam, ab6640), iNOS (Abcam, ab178945), and CD206 (Abcam, ab64693) were used in this process. The polarization states of macrophages on various modified surfaces were analysed via flow cytometry, Western blotting (WB), and RT-qPCR. Detailed descriptions of the methods used for WB and RT-qPCR are provided in a subsequent section. Here, we briefly introduce the flow cytometry procedure. The cells from each group were collected and incubated for 30 min with antibodies specific for F4/80 (Thermo, 11-4801-82), CD206 (Thermo, 17-2061-80), and CD86 (Thermo, 12-0862-81). After the samples were rinsed twice with PBS supplemented with 5% BSA, the cells were resuspended. Finally, the subsets of M1 and M2 macrophages within the different samples were identified using flow cytometry (Thermo Scientific, USA). Flow cytometry data were processed using FlowJo V10 software, and analyses were conducted in triplicate.

## Angiogenesis

This experiment is divided into two parts: 1) the indirect angiogenic effects of modified surfaces and 2) the direct angiogenic effects of modified surfaces. To investigate whether various modified surfaces can modulate the immune microenvironment to indirectly promote angiogenesis, supernatants were collected from BMDMs cultured on these surfaces. Next, the supernatants collected from the BMDMs and

fresh high-glucose DMEM were mixed in equal proportions to prepare conditioned medium (CM). HUVECs were plated in 6-well plates at a concentration of $2 \times 10^5$ cells per well and cultured in high-glucose DMEM supplemented with 10% FBS for 24 h. The next day, when the cells reached confluence, they were treated with 1 µg/mL mitomycin C (Sigma, M5353) for 1 h. After creating a uniform scratch in each well, the cells were washed three times with PBS, and the medium was replaced with CM. The scratch width was observed under a microscope (Zeiss, Germany) at 0 and 24 h. A migration assay was performed using a Transwell system (Corning, USA) to evaluate cell movement. HUVECs were plated at a concentration of $1 \times 10^4$ cells per well in the upper chamber, while CM from different groups was added to the lower chamber for a 24-h incubation. To evaluate cell migration, the cells were stained with 0.1% crystal violet solution (Solarbio, China). Ultimately, we performed an angiogenesis assay. Briefly, the ABW® Matrigel matrix was kept on ice and left to thaw overnight at 4 °C. After thawing, the matrix was mixed at a 1:1 ratio with fresh, serum-free DMEM. Fifty microlitres of the solution was added to a 96-well plate and placed in a cell culture incubator for 40 min to enable solidification. Next, HUVECs were introduced into a previously mentioned 96-well plate with 20,000 cells per well and incubated with CM from various modified surfaces. Tube formation in HUVECs was visualized at 4- and 24-- time points. Additionally, HUVECs were incubated in 6-well plates with CM from different groups. After 24 h, RNA and proteins were extracted for RT-qPCR and WB analysis.

Before confirming their direct angiogenic effects, the modified titanium surfaces were pretreated with a 1.5 µg/mL MMP-2 solution for 72 h to eliminate the influence of anti-inflammatory peptide sequences on the experiment. Subsequently, HUVECs were directly seeded onto the pretreated modified titanium surfaces and incubated for 24 h in DMEM supplemented with 10% FBS. The next day, after treatment with 1 µg/mL mitomycin C for 1 h, a uniform scratch was made on the modified titanium surface. Calcein–AM staining was performed at 0 and 24 h, and cell migration was analysed using an inverted fluorescence microscope. Since the titanium plates are impenetrable, we incorporated peptides from various modified titanium surfaces into DMEM and subsequently conducted migration and tube formation experiments using this peptide-enriched medium. A density of $1 \times 10^4$ cells per well was used to seed HUVECs in the upper chamber, with peptide-enriched medium from different groups in the lower chamber for a 24-h incubation, and cell migration was assessed using 0.1% crystal violet solution (Solarbio, China). The procedure for the tube formation assay was the same as above, except that the CM was replaced with peptide-enriched medium. Prior to examination, the samples were treated with calcein–AM and analysed using fluorescence microscopy. Finally, HUVECs were directly seeded onto titanium surfaces pretreated with MMP-2 for 72 h. RNA and proteins were then extracted for RT-qPCR and WB analysis.

## Osteogenesis

Similarly, experiments related to osteogenesis were conducted via two separate approaches: indirect and direct. First, to verify the indirect osteogenic effect of the surfaces, the supernatant from BMDMs incubated on various titanium surfaces was collected and mixed with fresh low-glucose DMEM at a 1:1 ratio to prepare macrophage CM. BMSCs were seeded in plates at a density of $2 \times 10^4$ cells per well and incubated for 36 h in low-glucose DMEM containing 10% FBS. The initial culture medium was replaced with CM containing osteogenic supplements, specifically 10 mM β-glycerophosphate, 0.1 µM dexamethasone, and 0.25 mM ascorbate. The media was changed every 2–3 days. On day 7, ALP staining was performed, and on day 21, ARS staining was carried out. The procedures followed the standard protocols provided with the ALP Staining Kit (Beyotime, C3206) and the ARS Staining Kit (Beyotime, C0148S). After the BMSCs were incubated for 48 h in CM containing osteogenic factors, immunofluorescence staining was

conducted to quantify the expression levels of OPN (Immunoway, YT3467) in the different groups. Simultaneously, RNA and proteins were extracted from different groups for RT-qPCR and WB analysis.

After the titanium plates were pretreated with 1.5 µg/mL MMP-2 solution for 72 h, BMSCs were directly seeded onto the pretreated titanium surfaces. The BMSCs were incubated in fresh low-glucose DMEM containing 10% FBS and osteogenic factors, and the medium was changed every 2–3 days. After 48 h, immunofluorescence staining was performed to assess the levels of COL-I (Abcam, ab270993) in the different groups. Simultaneously, RNA and proteins were extracted from different groups for RT-qPCR and WB analysis. Similarly, on days 7 and 21, ALP and ARS staining were performed directly on the incubated cells on the modified titanium surfaces.

## RT-qPCR and western blotting

Total RNA was extracted from cells subjected to various treatments using TRIzol™ reagent (Thermo Fisher, 15596026CN). Complementary DNA (cDNA) was synthesized from the extracted RNA through reverse transcription. The resulting cDNA was then amplified using a CFX96™ Real-Time PCR System (Bio-Rad, USA). The primers used for the target genes are detailed in Supplementary Table 1. Total protein was extracted after various treatments and then measured for quantification. The proteins were subjected to SDS–PAGE for 90 min at 110 V, followed by transfer onto PVDF membranes at 350 mA for 30 min. After blocking, the membranes were incubated overnight at 4 °C with primary antibodies targeting iNOS (Abcam, ab178945), CD206 (Abcam, ab64693), VEGF (Immunoway, YN5444), FGF-2 (Abcam, ab208687), COL-I (Abcam, ab270993), OPN (Immunoway, YT3467), and β-actin (Immunoway, YM3028). After thorough rinsing with TBST, the membranes were treated with appropriate secondary antibodies for one h.

## Micro-CT and biomechanical push-out experiments

Two months postsurgery, all harvested femurs underwent micro-CT scans using a SkyScan 1176 system (Belgium). The femur of each rat was imaged with a spatial resolution of 18 µm per layer, with the voltage and current set at 50 kV and 500 µA, respectively, and the rotational step was set at 0.7°. The micro-CT scan data were analysed using CTAn software, which was used to measure the bone mineral density (BMD, g/cm³), bone volume per tissue volume (BV/TV, %), trabecular number (Tb.N, 1/mm), and trabecular thickness (Tb.Th, µm) across various groups. Three-dimensional reconstruction of the CT data was performed to visually compare bone integration across the different groups. Five right femurs from each experimental group were subjected to biomechanical push-out testing. The maximum push-out force was evaluated utilizing the HY1080 material testing system (China). Before the mechanical tests, the femurs were fixed vertically to the base using dental cement. A force parallel to the femoral longitudinal axis was then applied to push the femurs downwards at a speed of 1 mm/min, with the maximum push-out force recorded during the process.

## Histological evaluation

Following biomechanical testing, the remaining five right femurs from each group, each containing a titanium rod, were dehydrated in 70% ethanol and subsequently used for hard tissue sectioning. The hard tissue sections containing titanium rods were subjected to van Gieson's (VG) staining to evaluate the variations in bone-to-implant contact (BIC) among the different groups. Concurrently, hard tissue sections labeled with calcein were examined to assess the differences in bone mineralization rates (MARs) between the groups. Ten remaining samples per group were harvested and subsequently immersed in a 10% ethylenediaminetetraacetic acid (EDTA) solution (RG-TG-01, Ketu Biotech) to facilitate gradual decalcification. The entire process was carried out in a constant-temperature shaking incubator at 37 °C for five weeks. Upon completion of the decalcification process, the titanium rods were meticulously extracted

from the femurs. The tissues were processed into paraffin sections for further study. Histological sections were meticulously prepared, and immunohistochemical staining was performed on these paraffin sections to measure the levels of COL-I and OPN. Collagen deposition around the titanium rods was also assessed via Masson staining. Additionally, the femurs of rats from each group, which were extracted on the fifth day postsurgery, were subjected to slow decalcification and processed into paraffin sections. These sections were initially stained with H&E to assess fibroproliferation around the titanium rods. Immunohistochemical staining was subsequently carried out to evaluate the differences in the expression of TNF-α (Servicebio, GB11188) and IL-10 (Servicebio, GB11534) between the groups. Finally, immunofluorescence staining was performed using CD68 (Abcam, ab201340) to label macrophages, CD86 (Immunoway, YT7823) as a marker of M1 polarization, and CD206 (Abcam, ab64693) as a marker of M2 polarization. All the staining processes were performed following the standard protocols outlined by the reagent suppliers.

### Transcriptome sequencing
We conducted two transcriptome sequencing analyses to explore the mechanisms of direct angiogenesis and direct osteogenesis on DOPA-P1@P2 surfaces. HUVECs and BMSCs were individually seeded onto $TiO_2$ and DOPA-P1@P2 surfaces that had been pretreated with MMP-2 and then incubated for two days. Following the extraction of total RNA, RNA purity and integrity were evaluated using an Agilent 2100 Bioanalyzer (Agilent Technologies, USA). Libraries were subsequently constructed using a VAHTS Universal V6 RNA-seq Library Prep Kit in accordance with the manufacturer's instructions. Transcriptome sequencing and analysis were carried out by OE Biotech Co., Ltd. (China).

### Statistical analysis
The data are expressed as the means ± standard deviations and were derived from a minimum of three replicates for each experimental sample. Differences between two groups were evaluated using the Student's t test, whereas one-way ANOVA was used for comparisons among multiple groups. Statistical significance was defined as $p < 0.05$ (*), and strong significance was defined as $p < 0.01$ (**). All the statistical evaluations were conducted with GraphPad Prism version 10.1.0.

### Reporting summary
Further information on research design is available in the Nature Portfolio Reporting Summary linked to this article.

## Data availability
The RNA-seq data generated in this study have been deposited in the SRA database under accession code PRJNA1166748 and PRJNA1166738 [https://www.ncbi.nlm.nih.gov/bioproject/PRJNA1166748;https://www.ncbi.nlm.nih.gov/bioproject/PRJNA1166738]. The source data underlying Figs. 1–8 and Supplementary Figs. 3, 5, 6–7, and 10–11 are provided as a Source data file. Source data are provided with this paper.

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

## Acknowledgements

This work was supported by the National Natural Science Foundation of China (82472503 to W.H., 82472437 to L.X., and 52203196 to X.N.), the Provincial Key R&D Programmes of Anhui (2022e07020016 to J.Z.) and the Hefei Natural Science Foundation (2022045 to J.Z.).

## Author contributions

W.Z. and W.H. conceived and designed the study. W.Z. and Y.L. conducted in vitro cell experiments, while W.Z. and X.N. performed materials characterization and animal experiments. C.Z. and L.X. designed and synthesized the peptides. W.Z., C.Z., L.X., J.Z., and W.H. contributed to manuscript drafting, and J.Z. and W.H. revised it. All authors discussed the results and reviewed the manuscript.

## Competing interests

The authors declare no competing interests.
