## [Transparent Peer Review file · Nature Communications]

Peptide-Based Inflammation-Responsive Implant Coating Sequentially Regulates Bone Regeneration to Enhance Interfacial Osseointegration

Corresponding Author: Professor Wei Huang

Version 0:

Reviewer comments:

Reviewer #1

(Remarks to the Author)

The authors present a comprehensive study to create a peptide-modified implant that is responsive to different stages of inflammation to promote bone regeneration and implant integration. The authors perform an extensive battery of characterization, in vitro assays, and in vivo studies. The paper is clearly written with robust statistical analysis. The writing style includes significant background in the Results before the data are presented, which seems unnecessary in that section. Overall, this is very nice work.

Major concerns

1. The title must be revised. As the sequential coating and responsiveness to desired stimuli is demonstrated, there is no evidence that the implant is “self-adaptive”. This would require additional studies in vivo to block MMPs and/or macrophages at the implant site and assess differences.
2. The abstract should describe briefly what K23, K15, and Y5 means. Even just defining the goal of the peptide is useful.
3. The use of osteogenic growth peptide, although efficacious, is not standard in the field. The authors should provide 1-2 sentences of context for why this peptide is the best choice. Furthermore, there are many other approaches that may promote angiogenesis and osteogenesis that are not referenced.
4. Experiments to study stability and release are in normal culture media at 37C. This is not physiologic and should be recognized as a limitation.
5. The use of RAW264.7 macrophages as a model are not ideal, as they are biased toward the M1 phenotype. The work would be strengthened by using bone marrow derived macrophages or a different macrophage cell line.
6. CCK-8 is for cell proliferation, not viability. These studies are not described accurately. There could be changes in metabolic activity leading to experimental changes.
7. Line 238: The authors mention that data are significant correlated with increased surface roughness, but no data are included to support the statement.
8. The continuous use of “timely transition” is vague and depends on the injury, animal model, and intervention. Is five days “timely” or arbitrarily chosen?
9. Data for HUVEC migration in the scratch wound fails to account for cell proliferation, which could be confounding factor.
10. Images in Figure 3 would be strengthened by co-staining for M1 and M2 markers so the reader can appreciate macrophage phenotype across the spectrum. It is important to realize that macrophages often express markers of both phenotypes and exist along a spectrum.
11. Figure 5: Data for conditioned media of macrophages alone are missing. As macrophages secrete potent cues, this control is necessary.

Minor concerns:

1. Please carefully check for grammatical errors. For example, line 67 has an unnecessary “of”.
2. Line 216: RAW264.7 cells are not abbreviated in this manner
3. Line 218: “cell types” may be better wording than “groups”
4. Line 314: this is a sentence fragment.
5. Define all abbreviations at first use. This is particularly true in the transcriptomics section of the manuscript.

Reviewer #2

(Remarks to the Author)

The manuscript entitled "Peptide-based Self-adaptive Implant Coating Sequentially Regulates Bone Regeneration to Enhance Interfacial Osseointegration" focuses on the development and study of a titanium implant coated with bioactive peptides for osseointegration. Results overall support that both P1 and P2 (separate and in combination) promote M2 macrophage polarization, while the combination of P1 and P2 results in the highest levels of angiogenesis and osteogenesis (section 2.5). The rationale behind the indirect measurements of angiogenesis and osteogenesis using conditioned media is not clear, I would recommend authors explain further how the peptides in the surfaces and the conditioned media interact to modulate angiogenesis and osseointegration. e.g. will the peptides be cleaved in presence of the conditioned media, released and act into HUVEC and BMSCs?

The work is a solid proof that a sequential approach linking the host immune response and regeneration, and mimicking the distinct stages of bone regeneration improves regenerative outcomes, concept that in other systems has increasingly been recognized. What is not clear, and should be explained further, given the wide audience of the journal, is why such approach is inspired in mussels.

The work done by the authors supports conclusions and claims, excepting the points mentioned above, that will likely only require further explanations to improve clarity. No flaws in analyses and interpretations were found. Methodologies meet publication standards in the field. Lack of important information was not found, with enough details to be reproduced by people in relevant fields of biomaterials and tissue engineering.

Other than the suggested revisions, I would also recommend to revise the abstract. In its current status, there are several things that are not detailed, including P1, P2, K23, K15, Y5 and so on. These questions are eventually answered when we read the rest of the article, but the abstract should be a stand-alone piece of information to summarize the work and capture the attention of the readers.

Page 31 931

Define what a robust osseointegration is.

Reviewer #3

(Remarks to the Author)

In this manuscript, the authors reported a self-adaptive multifunctional coating by using four peptides that correspond to the characteristic sequential targets within the inflammation stage, proliferation stage of bone regeneration after the implantation of the Ti based implants. Here are some questions for the authors' consideration.

- (1) For the ¹H NMR spectrum of (DOPA)₄-109 PEG₅-DBCO, the authors should provide the area integration of the characteristic peaks.
- (2) How did the authors screen the polypeptide sequence of K23, K15, Y5, and the MMP sensitive peptide, respectively? It seems that all of these peptides can be searched in previous reports. The authors should cite related references.
- (3) How about the surface grafting efficiency of P1 and P2?
- (4) A previous report showed that the K23 peptide (KFAKLAARLYRKALARQLGVAA) played its anti-inflammatory role after cell-penetration (doi.org/10.1016/j.nano.2012.09.003). How did the authors guaranteed the K23 peptides showed M1 to M2 regulation when the peptides are grafted in the surface of TiO₂?
- (5) Did the combination of K23 with K15 or Y5 affect the anti-inflammatory efficiency of K23 peptide?
- (6) In Line 178, the author wrote "The DOPA-P1@P2 group presented the smallest water contact angle (55.6°), indicating superior surface hydrophilicity." As there was no significant difference among DOPA-P1, DOPA-P2, and DOPA-P1@P2 groups, the use of "smallest" was not suitable.
- (7) Which pathway was more important for promoting angiogenesis and osteogenesis, regulation of cytokine secretion by M2 macrophages, or the direct regulation of grafting peptides.
- (8) After the transcriptome sequencing analysis, the authors should further prove the selected relative pathway by using RT-PCR, WB or cellular staining.
- (9) The authors pointed out the importance of "osteogenic-angiogenic coupling" in Line 320. However, there is no research about the interaction between HUVECs and MSCs.
- (10) The detachment of K23 peptide should be proved in vivo. The authors can label K23 peptide with fluorescence.

Version 1:

Reviewer comments:

Reviewer #1

(Remarks to the Author)

The authors have done a thorough job answering this reviewer's concerns, as well as those of the other reviewers, by performing additional studies and more fully characterizing the cell response, both in vitro and in vivo. The writing is improved. Congratulations on this interesting work.

Kent Leach
UC Davis

Reviewer #2

(Remarks to the Author)

The authors have satisfactorily addressed all comments and suggestions given by all reviewers. I was thinking the same regarding RAW 264.7 cells regarding being prone to polarization, I am glad the authors repeated the experiments on BMDMs. This and the other revisions have significantly increased the quality of the results and support of the hypotheses in the manuscript.

I believe the manuscript is now suitable for publication, with one minor suggestion: In the title for figure 2, cytocompatibility is more accurate than biocompatibility. I agree that in vivo experiments shown later demonstrate biocompatibility, but Figure 2 only focusses on in vitro cell culture testing.

Reviewer #3

(Remarks to the Author)

The authors have properly addressed all concerns with revisions. It is suggested to be accepted for publication.

Response to Reviewer #1:

Reviewer #1: The authors present a comprehensive study to create a peptide-modified implant that is responsive to different stages of inflammation to promote bone regeneration and implant integration. The authors perform an extensive battery of characterization, in vitro assays, and in vivo studies. The paper is clearly written with robust statistical analysis. The writing style includes significant background in the Results before the data are presented, which seems unnecessary in that section. Overall, this is very nice work.

Response: We are deeply grateful for your high recognition of our work. Your valuable feedback has been crucial in enhancing the quality of our research. In response to your suggestions, we have carefully revised the manuscript, with detailed changes outlined below. Additionally, regarding your comments on the writing style, we fully agree with your perspective. Certain textual content from the Results section has been extracted and reorganized into the Discussion to enhance clarity and improve reader comprehension (**Pages 16-20 Lines 433-562, marked in red**).

Revised version:

3. Discussion

Aseptic loosening due to insufficient osseointegration at the bone–implant interface remains a leading cause of prosthetic failure after joint arthroplasty³¹. The surface modification of bone implants to enhance bioactivity has shown promise in improving interfacial osseointegration³². Early efforts focused on enhancing the osteoinductive properties of implant surfaces^{33, 34}, but targeting osteogenesis alone was inadequate. Successful implant-mediated bone regeneration requires the coordinated interplay of immunomodulation, angiogenesis, and osteogenesis, which are temporally regulated. While emerging implant surfaces can modulate these processes simultaneously³⁵, insufficient attention to their temporal dynamics leads to disordered interactions, requiring further optimization. In this study, we developed an inflammation-responsive biomimetic titanium surface (DOPA-P1@P2) that effectively promotes bone regeneration by intentionally aligning with the physiological process. Previous studies reported that^{36, 37}, compared with TiO₂ control, the maximal push-out force increased by an average of 150%, bone volume to total

volume (BV/TV) increased by 120%, and bone-to-implant contact (BIC) increased by 300%. Remarkably, in this work, DOPA-P1@P2 achieved a 161% increase in the maximal push-out force, a 207% increase in BV/TV, and an extraordinary 1409% increase in BIC. These findings highlight notable advancements attributed to the novel design concept that aligns with the physiological processes of bone regeneration to enhance osseointegration.

Traditionally, surface modification of biomaterials relies on incorporating bioactive macromolecules, such as VEGF and BMP-2, or metal ions, such as Zn^{2+} and Mg^{2+} , to achieve functionalization³⁸⁻⁴¹. However, these macromolecules are difficult to chemically modify and prone to inactivation, and the toxicity and uncontrolled release of metal ions limit their application^{42, 43}. Recent advances in peptide synthesis have positioned peptides as promising alternatives for biomaterial surface modification, offering comparable bioactivity, increased safety, increased stability, and greater ease of chemical modification⁴⁴. Thus, in this work, we introduced four functional peptides to modify titanium-based surfaces. K23 (KAFAKLAARLYRKALARQLGVAA) is an anti-inflammatory, cell-penetrating peptide (CPP) that inhibits proinflammatory cytokines such as IL-1, IL-6, and TNF- α by targeting mitogen-activated protein kinase-activated protein kinase 2 (MK2)⁴⁵. It has been shown to reduce inflammation in a macrophage model with human monocytes⁴⁶, making it a suitable choice for its anti-inflammatory role in this study. K15 (KLTWQELYQLKYKGI), a synthetic 15-amino-acid polypeptide derived from the VEGF binding region with VEGF-like angiogenic activity, activates endothelial cells to mimic VEGF functions such as invasion, chemotaxis, and capillary sprouting⁴⁷, making it the ideal choice for promoting angiogenesis in this work. Y5 (YGFGG), the C-terminal fragment of osteogenic growth peptide (OGP), is the active component that enhances osteoblast proliferation and differentiation, was utilized in this study to promote osteogenesis²⁹. Macrophages within the inflammatory microenvironment regulate pathophysiological processes through the secretion of matrix metalloproteinases (MMPs), particularly MMP-2 and MMP-9⁴⁸. The peptide sequence PVGLIG, which is specifically cleaved by either MMP-2 or MMP-9, serves as an ideal linker between K23 and K15, as well as between K23 and Y5, enabling an inflammation-responsive effect during the first stage of bone regeneration²⁷.

To date, physical and chemical modification techniques have been widely employed in the field of biomaterial surface engineering. For example, methods such as layer-by-layer self-assembly, anodization, acid etching, and ion doping are commonly utilized to introduce bioactive molecules onto implant surfaces⁴⁹⁻⁵². However, these physical approaches often suffer from significant molecular leakage and face challenges in maintaining long-term bioactivity^{34, 53}, thereby limiting their practical application. On the other hand, conventional chemical modification methods often involve complex reactions and tedious surface treatment procedures, with limited controllability and operability, particularly in the development of multifunctional surface modifications⁵⁴. Thus, developing a simple and efficient surface modification technique capable of conjugating multiple bioactive molecules is critically important. Inspired by the adhesion mechanism of marine mussel foot proteins (Mfps), mussel-inspired bioinspired strategies have emerged as highly promising surface modification approaches, leveraging the repetitive catecholamine structure of 3,4-dihydroxy-L-phenylalanine (DOPA) in Mfps to achieve strong adhesion on nearly all types of surfaces^{36, 55}. Moreover, DOPA can effectively immobilize various bioactive molecules through covalent bonding while significantly enhancing the biocompatibility of material surfaces^{10, 34}. Given these advantages, mussel-inspired biomimetic strategies represent a great approach for surface modification in this study. However, when mussel-inspired peptides are used to modify titanium surfaces, the coordination between DOPA and TiO₂ may compromise the activity of biomolecules covalently linked to DOPA³⁷. Consequently, there is an urgent need to optimize the existing mussel-inspired surface modification strategies. Fortunately, the rise of bioorthogonal click chemistry has opened a new avenue, as its mild and direct reaction conditions, rapid reaction rates, and excellent specificity make this technique a current research hotspot in the field of implant surface modification^{56, 57}. The surface modification approach employed in this work combines mussel-inspired biomimetic strategies with bioorthogonal click chemistry, overcoming numerous drawbacks of traditional surface modification techniques. This method is straightforward, practical, and economical and holds promise for widespread application in settings requiring multifunctional surface modifications.

The primary prerequisite for the clinical application of implanted materials is their nontoxicity to the body⁵⁸. This study demonstrated the biocompatibility of DOPA-P1@P2 in three cell types (BMDMs, HUVECs, and BMSCs) and confirmed its *in vivo* safety for rat visceral organs, establishing a foundation for its practical application. Previous research has demonstrated that increased surface roughness enhances osteoblast adhesion and spreading, which facilitates osseointegration compared with smooth surfaces⁵⁹. Furthermore, enhancing the hydrophilicity of material surfaces promotes the maintenance of protein structure and function, enhances cellular adhesion, and improves biocompatibility⁶⁰. Compared with those of the TiO₂ control, the surface roughness and hydrophilicity of the biomimetic surfaces (DOPA-P1@P2) constructed in this work were significantly greater, facilitating osteoblast adhesion and spreading, which is critical for improving interfacial bone regeneration.

The initial stage of bone regeneration mediated by implant materials *in vivo* is characterized predominantly by an inflammatory response⁶¹. During this phase, a substantial haematoma forms on the implant surface, dominated by M1 macrophages, which release a plethora of inflammatory mediators, serving as a protective response against injury^{62, 63}. However, the resolution of acute inflammation during the early postimplantation period, particularly the transition from the proinflammatory M1 phenotype to the anti-inflammatory M2 phenotype before the proliferation stage, is a critical determinant of the fate of implant materials in the body⁶⁴. The prolonged dominance of M1 macrophages at the implant site leads to the sustained release of inflammatory mediators, which results in a shift from acute inflammation postimplantation to chronic inflammation. This triggers the foreign body response, with the recruitment of fibroblasts to encapsulate the implant material with a fibrous capsule, which impairs osseointegration and potentially leads to implant failure⁶⁵. Conversely, the transition from the M1 to the M2 macrophage phenotype before the proliferation stage can steer bone regeneration in the right direction. By secreting essential chemokines and growth factors, M2 macrophages enhance angiogenesis and the migration, homing, and osteogenic differentiation of BMSCs, thus optimizing conditions for subsequent stages of bone regeneration⁶⁶. According to the literature, the inflammatory response peaks within the first 24 hours of the inflammation stage, lasting approximately 7 days before transitioning to the proliferation stage⁶⁷. We selected the fifth day postimplantation (late inflammation stage) to

assess whether the material effectively promoted M2 polarization prior to the proliferation phase, which is consistent with previous reports^{36,37}. *In vivo* and *in vitro* experiments revealed that DOPA-P1@P2 effectively modulates the immune microenvironment and promotes macrophage M2 polarization, achieving 79.50% inhibition of M1 polarization, as shown by flow cytometry analysis, and demonstrated superior performance compared with related studies^{10,37}.

Angiogenesis and osteogenesis jointly constitute the proliferation stage of bone regeneration, which lasts 3 to 4 weeks^{15,68}. Neovascularization provides a vital connection between bone and the surrounding tissues, enabling the transport of oxygen, nutrients, and BMSCs to the injured area⁶⁹. The activation of the VEGF signalling pathway has been reported in the literature to effectively promote angiogenesis⁷⁰. The angiogenic peptide K15 used in this study is a VEGF-mimetic peptide that effectively promotes angiogenesis and activates the MAPK signalling pathway, which is consistent with previous reports^{71,72}. BMSCs around the injury site differentiate into osteoblasts in the local microenvironment, with osteogenic growth peptide (OGP) significantly increasing their proliferation and differentiation⁷³. Consistent with previous reports^{74,75}, the osteogenic peptide Y5 used in this study effectively promoted osteogenesis and activated the PI3K–Akt signalling pathway. However, angiogenesis and osteogenesis, both critical processes in bone regeneration, must be effectively integrated to achieve optimal interface osseointegration. In this study, the DOPA-P1@P2 group achieved superior osseointegration outcomes compared with the groups promoting angiogenesis (DOPA-P1) or osteogenesis (DOPA-P2) alone, likely due to the synergistic effects of angiogenesis and osteogenesis.

This study has several limitations, such as the inability of our experimental conditions to fully replicate the complexities of the physiological environment, including dynamic fluid flow, pH variations, and the influence of biological components (e.g., enzymes, proteins, and cells) on material behaviour. Additionally, the coupling mechanisms between angiogenesis and osteogenesis have not been explored further. In future studies, we will incorporate advanced materials (e.g., thermoresponsive or pH-responsive materials) to increase the responsiveness and precision of the coatings. Additionally, we will investigate the interaction

mechanisms between HUVECs and BMSCs to gain deeper insights into the physiological processes of bone regeneration.

Thanks again!

Major concerns

Comment 1: The title must be revised. As the sequential coating and responsiveness to desired stimuli is demonstrated, there is no evidence that the implant is “self-adaptive”. This would require additional studies in vivo to block MMPs and/or macrophages at the implant site and assess differences.

Response: Thank you for your constructive suggestions. After careful consideration, we decided to replace "Self-adaptive" with "Responsive." The revised title is as follows: Peptide-Based Inflammation-Responsive Implant Coating Sequentially Regulates Bone Regeneration to Enhance Interfacial Osseointegration (**Page 1, Lines 1-2, marked in red**).

Thanks again!

Comment 2: The abstract should describe briefly what K23, K15, and Y5 means. Even just defining the goal of the peptide is useful.

Response: Thank you very much for your valuable feedback. We fully agree with your perspective and have carefully revised the abstract to ensure it serves as a stand-alone summary that encapsulates our work and captures the readers' attention. The revised abstract is as follows:

Aseptic loosening remains the leading cause of bone prostheses failure, largely due to inadequate osseointegration caused by coatings that fail to align with the physiological process of bone regeneration. In this study, titanium surfaces were functionalized with the mussel-inspired peptide (DOPA)₄-OEG₅-DBCO. Anti-inflammatory (K23), angiogenic (K15), osteogenic (Y5), and inflammation-responsive (PVGLIG) sequences were then assembled into multifunctional peptides P1 (N₃-K15-PVGLIG-K23) and P2 (N₃-Y5-PVGLIG-K23), creating the inflammation-responsive coating DOPA-P1@P2 via click chemistry. DOPA-P1@P2 promoted bone regeneration through sequential regulation. In the initial stage, the outermost K23 layer induced M2 macrophage polarization, establishing a

pro-regenerative immune microenvironment. Subsequently, K15 and Y5 layers, exposed by the release of K23, enhanced angiogenesis and osteogenesis. In the final stage, DOPA-P1@P2 outperformed the TiO₂ control, showing a 161% increase in maximal push-out force, a 207% increase in bone volume fraction, and a 1409% increase in bone-to-implant contact. This study demonstrates that DOPA-P1@P2 significantly enhances interfacial osseointegration by sequentially regulating bone regeneration, providing new insights into the design of implant coatings (**Page 2, Lines 20-34, marked in red**).

Thanks again!

Comment 3: The use of osteogenic growth peptide, although efficacious, is not standard in the field. The authors should provide 1-2 sentences of context for why this peptide is the best choice. Furthermore, there are many other approaches that may promote angiogenesis and osteogenesis that are not referenced.

Response: Thank you for your valuable and insightful suggestion, which we are delighted to adopt. To this end, we have added the following content to the Discussion section (**Pages 16-17, Lines 452-458, marked in red**).

Traditionally, surface modification of biomaterials relies on incorporating bioactive macromolecules, such as VEGF and BMP-2, or metal ions, such as Zn²⁺ and Mg²⁺, to achieve functionalization¹⁻⁴. However, these macromolecules are difficult to chemically modify and prone to inactivation, and the toxicity and uncontrolled release of metal ions limit their application^{5, 6}. Recent advances in peptide synthesis have positioned peptides as promising alternatives for biomaterial surface modification, offering comparable bioactivity, increased safety, increased stability, and greater ease of chemical modification⁷.

In this work, we selected the YGFGG sequence, the minimal functional unit of osteogenic growth peptide (OGP) that retains its bioactive properties⁸. This sequence effectively regulates osteoblast proliferation, alkaline phosphatase activity, and matrix mineralization, making it an ideal candidate for our research⁹.

References:

1. Zhang, Bo et al. "An organic selenium and VEGF-conjugated bioinspired coating promotes vascular healing." *Biomaterials* vol. 287. (2022): 121654. doi:10.1016/j.biomaterials.2022.121654

2. Popova, Anastasiya D et al. “Osteoconductive, osteogenic, and antipathogenic plasma electrolytic oxidation coatings on titanium implants with BMP-2.” *ACS applied materials & interfaces* vol. 15,31. (2023): 37274-37289. doi:10.1021/acsami.3c08954
3. Li, Chunxu et al. “Continuously released Zn²⁺ in 3D-printed PLGA/β-TCP/Zn scaffolds for bone defect repair by improving osteoinductive and anti-inflammatory properties.” *Bioactive materials* vol. 24. (2022): 361-375. doi:10.1016/j.bioactmat.2022.12.015
4. Zhao, Yanbin et al. “pH/NIR-responsive and self-healing coatings with bacteria killing, osteogenesis, and angiogenesis performances on magnesium alloy.” *Biomaterials* vol. 301. (2023): 122237. doi:10.1016/j.biomaterials.2023.122237
5. Spicer, Christopher D, and Benjamin G Davis. “Selective chemical protein modification.” *Nature communications* vol. 5,4740. (2014). doi:10.1038/ncomms5740
6. Zhong, Qiang et al. “Prosthetic metals: release, metabolism and toxicity.” *International journal of nanomedicine* vol. 19. (2024): 5245-5267. doi:10.2147/IJN.S459255
7. Sharma, Anamika et al. “Liquid-phase peptide synthesis (LPPS): A third wave for the preparation of peptides.” *Chemical reviews* vol. 122,16 (2022): 13516-13546. doi:10.1021/acs.chemrev.2c00132
8. Chu, Yun Shin et al. “Combining Mg-Zn-Ca bulk metallic glass with a mesoporous silica nanocomposite for bone tissue engineering.” *Pharmaceutics* vol. 14,5. (2022): 1078. doi:10.3390/pharmaceutics14051078
9. Zhu, Haibao et al. “Controlled growth and differentiation of MSCs on grooved films assembled from monodisperse biological nanofibers with genetically tunable surface chemistries.” *Biomaterials* vol. 32,21 (2011): 4744-4752. doi:10.1016/j.biomaterials.2011.03.030

Thanks again!

Comment 4: Experiments to study stability and release are in normal culture media at 37C. This is not physiologic and should be recognized as a limitation.

Response: Thank you for your insightful comment. We acknowledge and sincerely accept that our experimental conditions do not fully capture the complexities of the physiological environment, such as dynamic fluid flow, pH variations, and the impact of biological components (e.g., enzymes, proteins, cells) on material behavior. However, given the challenges of directly monitoring peptide stability and release *in vivo*, PBS or culture media are commonly used as model systems in the literature for studying release behavior¹⁻³. These methods offer high reproducibility and experimental control. Following your suggestion, we have clarified this limitation in the manuscript (**Page 20, Lines 554–557, marked in red**) and plan to explore the *in vivo* stability and release of DOPA-P1@P2 in future studies.

Revised version:

This study has several limitations, such as the inability of our experimental conditions to fully replicate the complexities of the physiological environment, including dynamic fluid flow, pH variations, and the influence of biological components (e.g., enzymes, proteins, and cells) on material behaviour.

References:

1. Wang, Tao et al. "Engineering immunomodulatory and osteoinductive implant surfaces via mussel adhesion-mediated ion coordination and molecular clicking." *Nature communications* vol. 13,1. (2022): 160. doi:10.1038/s41467-021-27816-1
2. Zheng, Yanyan et al. "A programmed surface on polyetheretherketone for sequentially dictating osteoimmunomodulation and bone regeneration to achieve ameliorative osseointegration under osteoporotic conditions." *Bioactive materials* vol. 14. (2022): 364-376. doi:10.1016/j.bioactmat.2022.01.042
3. Li, Xiuying et al. "Microenvironmental enzyme-responsive methotrexate modified quercetin micelles for the treatment of rheumatoid arthritis." *International journal of nanomedicine* vol. 19. (2024): 3259-3273. doi:10.2147/IJN.S457004

Thanks again!

Comment 5: The use of RAW264.7 macrophages as a model are not ideal, as they are biased toward the M1 phenotype. The work would be strengthened by using bone marrow derived macrophages or a different macrophage cell line.

Response: Thank you for your valuable and insightful suggestion. We have reviewed the relevant literature¹⁻³ and fully agree with your perspective. We believe that bone marrow-derived macrophages (BMDMs) provide a more accurate model of macrophage behavior *in vivo*, enhancing the reliability of research findings. Accordingly, we repeated the relevant experiments using BMDMs, updated the results, and replaced the original **Figures 2, 3, 5, and S6** (as detailed below).

Figure 2: Biocompatibility assessment of modified surfaces. (A–C) Live/dead staining of BMDMs, HUVECs, and BMSCs on modified surfaces. (D) Cytoskeletal staining of BMSCs on modified surfaces using FITC-phalloidin and DAPI. (E–G) LDH cytotoxicity assays for BMDMs, HUVECs, and BMSCs cultured on modified surfaces. (H–M) CCK-8 assays for BMDMs, HUVECs, and BMSCs cultured on modified surfaces over 1 and 3 days. The data are presented as the mean \pm standard deviation (SD); $n = 5$ per group. Statistical analysis was performed by one-way ANOVA, and * $P < 0.05$ and ** $P < 0.01$ indicate statistical significance.

Figure 3: Regulation of macrophage polarization by modified surfaces *in vitro*. (A) Immunofluorescence staining evaluated macrophage polarization in BMDMs cultured on various modified surfaces, with the macrophage marker F4/80 visualized in green, the M1 marker iNOS in purple, the M2 marker CD206 in red, and nuclei in blue. (B–C) Quantitative analysis of immunofluorescence staining for target proteins. (D–E) ELISA quantified the secretion levels of the proinflammatory cytokine TNF- α and the anti-inflammatory cytokine IL-10. (F–H) Flow cytometry analyzed the expression of CD86 (M1 marker) and CD206 (M2 marker), and the corresponding quantitative analysis results. (I–K) WB detected the expression levels of iNOS and CD206 proteins and the corresponding quantitative analysis results. (L–O) RT-qPCR detected the gene expression levels of *Ccr7* (M1 marker), *Cd206* (M2 marker), *Vegf* (angiogenesis marker), and *Bmp2* (osteogenesis marker). The data are presented as the mean \pm standard deviation (SD); n = 3 or 5 per group. Statistical analysis was performed by one-way ANOVA, and *P < 0.05 and **P < 0.01 indicate statistical significance.

Figure 5: Evaluation of the indirect promotion of angiogenesis and osteogenesis by modified surfaces. (A) Experimental procedure flowchart. (B) Wound healing assays were performed using conditioned medium, and (I) quantitative analysis of the results was conducted. (C) Transwell assays and corresponding (J) quantitative results. (D) Tube formation assays and corresponding (K–L) quantitative results. (E–F) WB analysis of angiogenic (VEGF and FGF-2) and osteogenic (COL-1 and OPN) protein expression and (M–P) quantitative analysis results. (G–H) ALP and ARS staining and corresponding (Q–R) quantitative analysis results. The data are presented as the mean \pm standard deviation (SD) (n=5 per group). Statistical analysis was performed by one-way ANOVA, and *P < 0.05 and **P < 0.01 indicates statistical significance.

Figure S6: (A) Immunofluorescence staining evaluated the indirect osteogenic effects of modified surfaces, highlighting the cytoskeleton in green (F-actin), osteogenesis-related proteins in red (OPN), and nuclei in blue. (B) Quantitative results of immunofluorescence. (C–F) RT-qPCR assessed the expression of osteogenic (*Colla1* and *Opn*) and angiogenic (*VEGF* and *FGF2*) genes to evaluate the indirect osteogenic effects of modified surfaces. The data are presented as the mean \pm standard deviation (SD); n = 3 or 5 per group. Statistical analysis was performed by one-way ANOVA, and *P < 0.05 and **P < 0.01 indicate statistical significance.

References:

1. Zhao, Yan-Long et al. "Comparison of the characteristics of macrophages derived from murine spleen, peritoneal cavity, and bone marrow." *Journal of zhejiang university. science. B* vol. 18,12. (2017): 1055-1063. doi:10.1631/jzus.B1700003
2. Sun, Jie et al. "Bio-clickable mussel-inspired peptides improve titanium-based material osseointegration synergistically with immunopolarization-regulation." *Bioactive materials* vol. 9. (2021): 1-14. doi:10.1016/j.bioactmat.2021.10.003
3. Bai, Jiaxiang et al. "Biomimetic osteogenic peptide with mussel adhesion and osteoimmunomodulatory functions to ameliorate interfacial osseointegration under chronic inflammation." *Biomaterials* vol. 255. (2020): 120197. doi:10.1016/j.biomaterials.2020.120197

Thanks again!

Comment 6: CCK-8 is for cell proliferation, not viability. These studies are not described accurately. There could be changes in metabolic activity leading to experimental changes.

Response: We sincerely thank you for your thoughtful review of our manuscript. Your valuable comments have not only helped improve the quality of our research but have also provided us with a learning opportunity. We acknowledge that our initial understanding of the purpose of the CCK-8 assay was incomplete, and we regret overlooking the potential

influence of metabolic activity. To more accurately present the experimental results, we have revised the y-axis label of the histograms in Figure 2 to "Absorbance (OD₄₅₀)" and carefully updated the corresponding description of the results (**Page 7, Lines 185–191, highlighted in red**). The revised results description and figure are as follows:

Additionally, the absorbance at 450 nm for three different cell types on various modified surfaces was measured at 24 and 72 hours using CCK-8 assays (Figure 2H–M). The findings indicated that, by day 3, the absorbance at 450 nm on the four modified surfaces (DOPA, DOPA-P1, DOPA-P2, and DOPA-P1@P2) was considerably higher compared to the TiO₂ control surface, suggesting that peptide modifications of titanium surfaces significantly promote cell proliferation.

Figure 2: Biocompatibility assessment of modified surfaces. (A–C) Live/dead staining of BMDMs, HUVECs, and BMSCs on modified surfaces. (D) Cytoskeletal staining of BMSCs on modified surfaces using FITC-phalloidin and DAPI. (E–G) LDH cytotoxicity assays for BMDMs, HUVECs, and BMSCs cultured on modified surfaces. (H–M) CCK-8 assays for BMDMs, HUVECs, and BMSCs cultured on modified surfaces over 1 and 3 days. The data are presented as the mean \pm standard deviation (SD); n = 5 per group. Statistical analysis was performed by one-way ANOVA, and *P < 0.05 and **P < 0.01 indicate statistical significance.

Thanks again!

Comment 7: Line 238: The authors mention that data are significant correlated with increased surface roughness, but no data are included to support the statement.

Response: We sincerely thank you for your thoughtful review and valuable suggestion. We have carefully revised this sentence to make it more precise (**Page 8, Lines 197–200, highlighted in red**), as follows:

The materials characterization results suggest that the DOPA-P1@P2 surfaces, compared to the TiO₂ group, exhibit significantly increased roughness and hydrophilicity (Figure 1K–N), which may explain the enhanced adhesion and spreading of BMSCs.

Thanks again!

Comment 8: The continuous use of “timely transition” is vague and depends on the injury, animal model, and intervention. Is five days “timely” or arbitrarily chosen?

Response: Thank you for your insightful and valuable comment. We sincerely apologize for using the vague term "timely transition" in our manuscript and have revised the relevant section (**Page 19, Lines 517–527, highlighted in red**) to ensure greater precision. Implant-mediated bone regeneration is a multifaceted physiological process that involves various cells and factors and comprises three stages, namely inflammation, proliferation and remodelling, over temporally consecutive frames^{1, 2}. According to the literature, the inflammatory response peaks within the first 24 hours of the inflammation stage, lasting about 7 days before transitioning to the proliferation stage³. The transition of macrophages from the pro-inflammatory M1 phenotype to the anti-inflammatory M2 phenotype prior to the proliferation stage plays a pivotal role in angiogenesis and osteogenesis. In this study, the fifth day post-implantation was not arbitrarily chosen but deliberately selected at the late stage of inflammation to evaluate whether the material effectively promotes M2 polarization before the proliferation stage. This time point aligns with previous reports, in which macrophage polarization phenotypes are typically assessed on days 4 to 5 post-implantation^{4,5}.

Revised version:

However, the resolution of acute inflammation during the early postimplantation period, particularly the transition from the proinflammatory M1 phenotype to the anti-inflammatory M2 phenotype before the proliferation stage, is a critical determinant of the fate of implant

materials in the body⁶⁴. The prolonged dominance of M1 macrophages at the implant site leads to the sustained release of inflammatory mediators, which results in a shift from acute inflammation postimplantation to chronic inflammation. This triggers the foreign body response, with the recruitment of fibroblasts to encapsulate the implant material with a fibrous capsule, which impairs osseointegration and potentially leads to implant failure⁶⁵. Conversely, the transition from the M1 to the M2 macrophage phenotype before the proliferation stage can steer bone regeneration in the right direction.

References:

1. Zhou, Xingzhi et al. "Spatiotemporal regulation of angiogenesis/osteogenesis emulating natural bone healing cascade for vascularized bone formation." *Journal of nanobiotechnology* vol. 19,1. (2021): 420. doi:10.1186/s12951-021-01173-z
2. Wu, Minhao et al. "Smart-responsive multifunctional therapeutic system for improved regenerative microenvironment and accelerated bone regeneration via mild photothermal therapy." *Advanced science (Weinheim, Baden-Wurttemberg, Germany)* vol. 11,2. (2024): e2304641. doi:10.1002/advs.202304641
3. Cho, Tae-Joon et al. "Differential temporal expression of members of the transforming growth factor beta superfamily during murine fracture healing." *Journal of bone and mineral research : the official journal of the American Society for Bone and Mineral Research* vol. 17,3. (2002): 513-520. doi:10.1359/jbmr.2002.17.3.513
4. Wang, Tao et al. "Engineering immunomodulatory and osteoinductive implant surfaces via mussel adhesion-mediated ion coordination and molecular clicking." *Nature communications* vol. 13,1. (2022)160. doi:10.1038/s41467-021-27816-1
5. Sun, Jie et al. "Bio-clickable mussel-inspired peptides improve titanium-based material osseointegration synergistically with immunopolarization-regulation." *Bioactive materials* vol. 9. (2021): 1-14. doi:10.1016/j.bioactmat.2021.10.003

Thanks again!

Comment 9: Data for HUVEC migration in the scratch wound fails to account for cell proliferation, which could be confounding factor.

Response: We sincerely appreciate your valuable feedback and fully agree with your perspective. Although serum-free medium was used in our scratch assay to reduce the impact of cell proliferation, it remained a potential confounding factor in our observations. To address this issue, we redesigned and conducted experiments in which cells were treated with

mitomycin C (1 $\mu\text{g/ml}$) for 1 hour prior to scratching, aiming to minimize the influence of cell proliferation on the results. The updated experimental results are presented in **Figures 5 and 6**.

Figure 5: Evaluation of the indirect promotion of angiogenesis and osteogenesis by modified surfaces. (A) Experimental procedure flowchart. (B) Wound healing assays were performed using conditioned medium, and (I) quantitative analysis of the results was conducted. (C) Transwell assays and corresponding (J) quantitative results. (D) Tube formation assays and corresponding (K–L) quantitative results. (E–F) WB analysis of angiogenic (VEGF and FGF-2) and osteogenic (COL-I and OPN) protein expression and (M–P) quantitative analysis results. (G–H) ALP and ARS staining and corresponding (Q–R) quantitative analysis results. The data are presented as the mean \pm standard deviation (SD) ($n=5$ per group). Statistical analysis was performed by one-way ANOVA, and * $P < 0.05$ and ** $P < 0.01$ indicates statistical significance.

Figure 6: Evaluation of the direct promotion of angiogenesis and osteogenesis by modified surfaces. (A) Experimental procedure flowchart. (B) Wound healing assays were conducted directly on the modified titanium surfaces, along with the corresponding (I) quantitative analysis results. (C) Transwell assays and corresponding (J) quantitative results. (D) Tube formation assays and corresponding (K–L) quantitative results. (E–F) WB analysis of angiogenic (VEGF and FGF2) and osteogenic (COL-I and OPN) protein expression and (M–P) quantitative analysis results. (G–H) ALP and ARS staining of modified titanium surfaces, along with the corresponding (Q–R) quantitative analysis results. The data are presented as the mean ± standard deviation (SD) (n=3 per group). Statistical analysis was performed by one-way ANOVA (*P < 0.05 and **P < 0.01 vs. the TiO₂ group or comparisons between combined groups indicated in the figure; #P < 0.05 and ##P < 0.01 vs. the DOPA group; ^P < 0.05 and ^^P < 0.01 vs. the DOPA-P2 group). P < 0.05 indicate statistical significance.

Thanks again!

Comment 10: Images in Figure 3 would be strengthened by co-staining for M1 and M2 markers so the reader can appreciate macrophage phenotype across the spectrum. It is important to realize that macrophages often express markers of both phenotypes and exist along a spectrum.

Response: We greatly appreciate your valuable comments and fully agree with your perspective. To better illustrate the effects of the material on macrophage polarization phenotypes, we performed co-staining experiments using the M1 marker iNOS and the M2 marker CD206. The corresponding results have been updated in Figure 3.

Figure 3: Regulation of macrophage polarization by modified surfaces *in vitro*. (A) Immunofluorescence staining evaluated macrophage polarization in BMDMs cultured on various modified surfaces, with the macrophage marker F4/80 visualized in green, the M1 marker iNOS in purple, the M2 marker CD206 in red, and nuclei in blue. (B–C) Quantitative analysis of immunofluorescence staining for target proteins. (D–E) ELISA quantified the secretion levels of

the proinflammatory cytokine TNF- α and the anti-inflammatory cytokine IL-10. **(F–H)** Flow cytometry analyzed the expression of CD86 (M1 marker) and CD206 (M2 marker), and the corresponding quantitative analysis results. **(I–K)** WB detected the expression levels of iNOS and CD206 proteins and the corresponding quantitative analysis results. **(L–O)** RT-qPCR detected the gene expression levels of *Ccr7* (M1 marker), *Cd206* (M2 marker), *Vegf* (angiogenesis marker), and *Bmp2* (osteogenesis marker). The data are presented as the mean \pm standard deviation (SD); n = 3 or 5 per group. Statistical analysis was performed by one-way ANOVA, and *P < 0.05 and **P < 0.01 indicate statistical significance.

Thanks again!

Comment 11: Figure 5: Data for conditioned media of macrophages alone are missing. As macrophages secrete potent cues, this control is necessary.

Response: We sincerely appreciate your valuable feedback, which has been crucial in enhancing the reliability of our research. In response to your suggestion, we have included a control group (conditioned medium of macrophages alone, designated as the CM group) and repeated all the experiments shown in **Figure 5**. The corresponding results have been updated accordingly in the revised manuscript **(Pages 10–11, Lines 260–302, highlighted in red)**. The updated results are consistent with the previous findings. Furthermore, we observed no statistically significant differences between the CM group and the TiO₂ group in promoting angiogenesis and osteogenesis.

Figure 5: Evaluation of the indirect promotion of angiogenesis and osteogenesis by modified surfaces. (A) Experimental procedure flowchart. (B) Wound healing assays were performed using conditioned medium, and (I) quantitative analysis of the results was conducted. (C) Transwell assays and corresponding (J) quantitative results. (D) Tube formation assays and corresponding (K–L) quantitative results. (E–F) WB analysis of angiogenic (VEGF and FGF-2) and osteogenic (COL-1 and OPN) protein expression and (M–P) quantitative analysis results. (G–H) ALP and ARS staining and corresponding (Q–R) quantitative analysis results. The data are presented as the mean \pm standard deviation (SD) (n=5 per group). Statistical analysis was performed by one-way ANOVA, and *P < 0.05 and **P < 0.01 indicates statistical significance.

Thanks again!

Minor concerns:

Comment 1: Please carefully check for grammatical errors. For example, line 67 has an unnecessary “of”.

Response: We greatly appreciate your meticulous review. We have corrected this error (**Page 3, Lines 68–70, highlighted in red**) and carefully re-checked the manuscript. Additionally, we enlisted the help of a professional editing service to minimize language errors.

Revised version:

Prolonged M1 polarization induces transformation of acute inflammation into a chronic state with the subsequent formation of peri-implant fibrous capsule leading to implant failures.

Thanks again!

Comment 2: Line 216: RAW264.7 cells are not abbreviated in this manner

Response: We greatly appreciate your careful review. This error (**Page 7, Lines 175–178, highlighted in red**) has been corrected, and in response to your Comment 5, RAW264.7 cells have been replaced with bone marrow-derived macrophages (BMDMs).

Revised version:

In this study, bone marrow-derived macrophages (BMDMs), human umbilical vein endothelial cells (HUVECs), and bone marrow-derived mesenchymal stem cells (BMSCs) were used as research subjects to evaluate biocompatibility of DOPA-P1@P2 across different cell types.

Thanks again!

Comment 3: Line 218: “cell types” may be better wording than “groups”

Response: Thank you for your valuable comment. We completely agree with your suggestion and have replaced “groups” with “cell types” to improve the precision of the expression (**Page 7, Lines 175–178, highlighted in red**).

Revised version:

In this study, bone marrow-derived macrophages (BMDMs), human umbilical vein endothelial cells (HUVECs), and bone marrow-derived mesenchymal stem cells (BMSCs)

were used as research subjects to evaluate biocompatibility of DOPA-P1@P2 across different cell types.

Thanks again!

Comment 4: Line 314: this is a sentence fragment.

Response: We sincerely appreciate your meticulous review and valuable suggestion. The error has been corrected (**Page 20, Lines 541–542, highlighted in red**), and we have conducted a thorough re-examination of the manuscript to avoid the recurrence of such issues.

Revised version:

The activation of the VEGF signalling pathway has been reported in the literature to effectively promote angiogenesis.

Thanks again!

Comment 5: Define all abbreviations at first use. This is particularly true in the transcriptomics section of the manuscript.

Response: Thank you for your valuable comment. We have carefully reviewed the entire manuscript and ensured that all abbreviations are fully defined at their first mention. This has been addressed with special attention to the transcriptomics section, where we have now clarified all technical terms and abbreviations to enhance the clarity and comprehensibility for readers.

Thanks again!

Response to Reviewer #2:

Comment 1: The manuscript entitled "Peptide-based Self-adaptive Implant Coating Sequentially Regulates Bone Regeneration to Enhance Interfacial Osseointegration" focuses on the development and study of a titanium implant coated with bioactive peptides for osseointegration. Results overall support that both P1 and P2 (separate and in combination) promote M2 macrophage polarization, while the combination of P1 and P2 results in the highest levels of angiogenesis and osteogenesis (section 2.5). The rationale behind the indirect measurements of angiogenesis and osteogenesis using conditioned media is not clear, I would recommend authors explain further how the peptides in the surfaces and the conditioned media interact to modulate angiogenesis and osseointegration. e.g. will the peptides be cleaved in presence of the conditioned media, released and act into HUVEC and BMSCs?

Response: We greatly appreciate your precise summary of our study and the thought-provoking question. To address your concerns, we employed CY5.5-labeled K23 and conducted macrophage polarization experiments on the DOPA-P1@P2 titanium surface. The results demonstrated that K23 can be released from the titanium surface and effectively penetrate into BMDMs (SFigure 2). Previous studies have confirmed that K23 is a cell-penetrating peptide capable of effectively promoting M2 macrophage polarization^{1, 2}. Therefore, we hypothesize that the indirect promotion of angiogenesis and osteogenesis may result from the release of pro-angiogenic and pro-osteogenic factors by polarized M2 macrophages. To test this hypothesis, we collected conditioned media from different experimental groups and performed ELISA analysis, which revealed significantly elevated levels of pro-angiogenic factor VEGF and pro-osteogenic factor BMP-2 in media from P1- or P2-modified titanium surfaces compared to TiO₂ group. Similar results were also validated at the gene level (as shown in Figure S5). These findings indicate that the mechanism involves promoting M2 macrophage polarization, thereby creating a favorable immune microenvironment that induces the release of factors beneficial for tissue repair, ultimately leading to the indirect promotion of angiogenesis and osteogenesis.

Figure 2 Cy5.5-labeled K23 effectively penetrated into BMDMs. Red: Cy5.5-labeled K23; Green: macrophage marker F4/80; Blue: nuclei.

Figure S5: (A-B) ELISA quantified the secretion levels of the angiogenesis-related protein VEGF and the osteogenesis-related protein BMP-2. (C-D) RT-qPCR detected the gene expression levels of *Vegf* (angiogenesis marker) and *Bmp2* (osteogenesis marker). The data are presented as the mean \pm standard deviation (SD); $n = 3$ or 5 per group. Statistical analysis was performed by one-way ANOVA, and $*P < 0.05$ and $**P < 0.01$ indicate statistical significance.

References:

1. He, Zhijiang et al. "An anti-inflammatory peptide and brain-derived neurotrophic factor-modified hyaluronan-methylcellulose hydrogel promotes nerve regeneration in rats with spinal cord injury." *International journal of nanomedicine* vol. 14. (2019): 721-732. doi:10.2147/IJN.S187854
2. Zhou, Kai et al. "Activated macrophage membrane-coated nanoparticles relieve osteoarthritis-induced synovitis and joint damage." *Biomaterials* vol. 295. (2023): 122036. doi:10.1016/j.biomaterials.2023.122036

Thanks again!

Comment 2: The work is a solid proof that a sequential approach linking the host immune response and regeneration, and mimicking the distinct stages of bone regeneration improves regenerative outcomes, concept that in other systems has increasingly been recognized. What is not clear, and should be explained further, given the wide audience of the journal, is why such approach is inspired in mussels.

Response: We sincerely appreciate your recognition of our work and your valuable comment. We would like to clarify that the inspiration for this sequential regulatory design is not derived from mussels but rather from a deep understanding of the pathological processes underlying bone regeneration. By analyzing the distinct characteristics of each stage of bone regeneration, we developed specific functional peptides tailored to meet the corresponding physiological demands. However, the effective modification of these peptides onto titanium-based surfaces remains a critical challenge. To date, physical and chemical modification techniques have been widely employed in the field of biomaterial surface engineering. For example, methods such as layer-by-layer self-assembly, anodization, acid etching, and ion doping are commonly utilized to introduce bioactive molecules onto implant surfaces¹⁻⁴. However, these physical approaches often suffer from significant molecular leakage and face challenges in maintaining long-term bioactivity^{5, 6}, thereby limiting their practical application. On the other hand, conventional chemical modification methods often involve complex reactions and tedious surface treatment procedures, with limited controllability and operability, particularly in the development of multifunctional surface modifications⁷. Thus, developing a simple and efficient surface modification technique capable of conjugating multiple bioactive molecules is critically important. Inspired by the adhesion mechanism of marine mussel foot proteins (Mfps), mussel-inspired bioinspired strategies have emerged as highly promising surface modification approaches, leveraging the repetitive catecholamine structure of 3,4-dihydroxy-L-phenylalanine (DOPA) in Mfps to achieve strong adhesion on nearly all types of surfaces^{8,9}. Moreover, DOPA can effectively immobilize various bioactive molecules through covalent bonding while significantly enhancing the biocompatibility of material surfaces^{5, 10}. Given these advantages, mussel-inspired biomimetic strategies represent a great approach for surface modification in this study.

To enhance readers' understanding of the rationale behind our selection of mussel-inspired biomimetic modification strategies, we have incorporated the aforementioned red-marked content into the Discussion section of the manuscript (**Pages 17–18, Lines 475–492, marked in red**).

References:

1. Hasani-Sadrabadi, M M et al. “Antibacterial and osteoinductive implant surface using layer-by-layer assembly.” *Journal of dental research* vol. 100,10. (2021): 1161-1168. doi:10.1177/00220345211029185
2. Sjöström, Terje et al. “Novel anodization technique using a block copolymer template for nanopatterning of titanium implant surfaces.” *ACS applied materials & interfaces* vol. 4,11. (2012): 6354-6361. doi:10.1021/am301987e
3. Qin, Wen et al. “Osseointegration and biosafety of graphene oxide wrapped porous CF/PEEK composites as implantable materials: The role of surface structure and chemistry.” *Dental materials : official publication of the Academy of Dental Materials* vol. 36,10. (2020): 1289-1302. doi:10.1016/j.dental.2020.06.004
4. Guo, Shuo et al. “A vessel subtype beneficial for osteogenesis enhanced by strontium-doped sodium titanate nanorods by modulating macrophage polarization.” *Journal of materials chemistry. B* vol. 8,28. (2020): 6048-6058. doi:10.1039/d0tb00282h
5. Pan, Guoqing et al. “Biomimetic design of mussel-derived bioactive peptides for dual-functionalization of titanium-based biomaterials.” *Journal of the American Chemical Society* vol. 138,45. (2016): 15078-15086. doi:10.1021/jacs.6b09770
6. Li, Jun et al. “Balancing bacteria-osteoblast competition through selective physical puncture and biofunctionalization of ZnO/polydopamine/arginine-glycine-aspartic acid-cysteine nanorods.” *ACS nano* vol. 11,11. (2017): 11250-11263. doi:10.1021/acsnano.7b05620
7. Mou, Xiaohui et al. “Mussel-inspired and bioclickable peptide engineered surface to combat thrombosis and infection.” *Research (Washington, D.C.)* vol. 2022. (2022): 9780879. doi:10.34133/2022/9780879
8. Lee, Haeshin et al. “A reversible wet/dry adhesive inspired by mussels and geckos.” *Nature* vol. 448,7151. (2007): 338-341. doi:10.1038/nature05968
9. Wang, Tao et al. “Engineering immunomodulatory and osteoinductive implant surfaces via mussel adhesion-mediated ion coordination and molecular clicking.” *Nature communications* vol. 13,1. (2022): 160. doi:10.1038/s41467-021-27816-1
10. Bai, Jiayang et al. “Biomimetic osteogenic peptide with mussel adhesion and osteoimmunomodulatory functions to ameliorate interfacial osseointegration under chronic inflammation.” *Biomaterials* vol. 255. (2020): 120197. doi:10.1016/j.biomaterials.2020.120197

Thanks again!

Comment 3: The work done by the authors supports conclusions and claims, excepting the points mentioned above, that will likely only require further explanations to improve clarity. No flaws in analyses and interpretations were found. Methodologies meet publication standards in the field. Lack of important information was not found, with enough details to be reproduced by people in relevant fields of biomaterials and tissue engineering.

Response: We sincerely appreciate your high recognition of our work and your positive comments. We have provided detailed explanations and included additional experimental results in the revised manuscript to address the concerns raised in Comment 1 and Comment 2. Following your suggestions, we have rewritten the abstract (**Page 2, Lines 20–34, marked in red**) and corrected the phrasing (**Page 34, Line 1017, marked in red**). Thank you for your valuable comments, which have been instrumental in enhancing the quality of our manuscript and strengthening the scientific rigor of our study.

Thanks again!

Comment 4: Other than the suggested revisions, I would also recommend to revise the abstract. In its current status, there are several things that are not detailed, including P1, P2, K23, K15, Y5 and so on. These questions are eventually answered when we read the rest of the article, but the abstract should be a stand-alone piece of information to summarize the work and capture the attention of the readers.

Response: Thank you very much for your valuable feedback. We fully agree with your perspective and have carefully revised the abstract to ensure it serves as a stand-alone summary that encapsulates our work and captures the readers' attention. The revised abstract is as follows:

Aseptic loosening remains the leading cause of bone prostheses failure, largely due to inadequate osseointegration caused by coatings that fail to align with the physiological process of bone regeneration. In this study, titanium surfaces were functionalized with the mussel-inspired peptide (DOPA)₄-OEG₅-DBCO. Anti-inflammatory (K23), angiogenic (K15), osteogenic (Y5), and inflammation-responsive (PVGLIG) sequences were then assembled into multifunctional peptides P1 (N₃-K15-PVGLIG-K23) and P2 (N₃-Y5-PVGLIG-K23), creating the inflammation-responsive coating DOPA-P1@P2 via

click chemistry. DOPA-P1@P2 promoted bone regeneration through sequential regulation. In the initial stage, the outermost K23 layer induced M2 macrophage polarization, establishing a pro-regenerative immune microenvironment. Subsequently, K15 and Y5 layers, exposed by the release of K23, enhanced angiogenesis and osteogenesis. In the final stage, DOPA-P1@P2 outperformed the TiO₂ control, showing a 161% increase in maximal push-out force, a 207% increase in bone volume fraction, and a 1409% increase in bone-to-implant contact. This study demonstrates that DOPA-P1@P2 significantly enhances interfacial osseointegration by sequentially regulating bone regeneration, providing new insights into the design of implant coatings (**Page 2, Lines 20-34, marked in red**).

Thanks again!

Comment 5: Page 31 931 Define what a robust osseointegration is.

Response: We sincerely apologize for any confusion caused by our choice of wording. What we intended to convey is that, in the third stage of bone regeneration, the implant interface achieves satisfactory osseointegration, characterized by the direct deposition of a substantial amount of regenerated bone on the implant surface and the consequent formation of a solid bone-to-implant anchorage without fibrous interference. After careful consideration, we have replaced "robust osseointegration" with "satisfactory osseointegration" (**Page 34, Line 1017, marked in red**) to better reflect our intended meaning.

Revised version:

In the third stage of bone regeneration, the implant interface achieves satisfactory osseointegration.

Thanks again!

Response to Reviewer #3:

Reviewer #3: In this manuscript, the authors reported a self-adaptive multifunctional coating by using four peptides that correspond to the characteristic sequential targets within the inflammation stage, proliferation stage of bone regeneration after the implantation of the Ti based implants. Here are some questions for the authors' consideration.

Response: We sincerely appreciate your highly precise summary of our research. Your valuable feedback has been crucial in enhancing the quality of our research. In response to your comments, we have carefully revised the manuscript, with detailed changes outlined below.

Thanks again!

Comment 1: For the ^1H NMR spectrum of $(\text{DOPA})_4\text{-109 PEG}_5\text{-DBCO}$, the authors should provide the area integration of the characteristic peaks.

Response: We sincerely appreciate your valuable suggestion. In response, we have provided the area integration of the characteristic peaks in Figure 1J, with detailed explanations added in the figure legend (**Page 35, Lines 1025–1027, highlighted in red**).

Revised version:

Figure 1 (J) The ^1H NMR spectrum of $(\text{DOPA})_4\text{-OEG}_5\text{-DBCO}$ displays characteristic peaks (a–f) with corresponding area integrations highlighted in blue. The DOPA unit (a) is marked by a peak at 8.64 ppm.

Thanks again!

Comment 2: How did the authors screen the polypeptide sequence of K23, K15, Y5, and the MMP sensitive peptide, respectively? It seems that all of these peptides can be searched in previous reports. The authors should cite related references.

Response: We greatly appreciate your valuable comment. The selection of K23, K15, Y5, and the MMP sensitive peptide (PVGLIG) was based on previous studies, which identified their functional properties and potential applications through a thorough literature review. Following your suggestion, we have clarified the rationale for selecting these peptides and added the relevant references in the Discussion section of the revised manuscript (**Page 17, Lines 459–474, highlighted in red**).

Revised version:

K23 (KAFAKLAARLYRKALARQLGVAA) is an anti-inflammatory, cell-penetrating peptide (CPP) that inhibits proinflammatory cytokines such as IL-1, IL-6, and TNF- α by targeting mitogen-activated protein kinase-activated protein kinase 2 (MK2)¹. It has been shown to reduce inflammation in a macrophage model with human monocytes², making it a suitable choice for its anti-inflammatory role in this study. K15 (KLTWQELYQLKYKGI), a synthetic 15-amino-acid polypeptide derived from the VEGF binding region with VEGF-like angiogenic activity, activates endothelial cells to mimic VEGF functions such as invasion, chemotaxis, and capillary sprouting³, making it the ideal choice for promoting angiogenesis in this work. Y5 (YGFGG), the C-terminal fragment of osteogenic growth peptide (OGP), is the active component that enhances osteoblast proliferation and differentiation, was utilized in this study to promote osteogenesis⁴. Macrophages within the inflammatory microenvironment regulate pathophysiological processes through the secretion of matrix metalloproteinases (MMPs), particularly MMP-2 and MMP-9⁵. The peptide sequence PVGLIG, which is specifically cleaved by either MMP-2 or MMP-9, serves as an ideal linker between K23 and K15, as well as between K23 and Y5, enabling an inflammation-responsive effect during the first stage of bone regeneration⁶.

References:

1. He, Zhijiang et al. "An anti-inflammatory peptide and brain-derived neurotrophic factor-modified hyaluronan-methylcellulose hydrogel promotes nerve regeneration in rats with spinal cord injury." *International journal of nanomedicine* vol. 14. (2019): 721-732. doi:10.2147/IJN.S187854
2. Bartlett, Rush L 2nd et al. "Cell-penetrating peptides released from thermosensitive nanoparticles suppress pro-inflammatory cytokine response by specifically targeting inflamed cartilage explants." *Nanomedicine : nanotechnology, biology, and medicine* vol. 9,3. (2013): 419-427. doi:10.1016/j.nano.2012.09.003
3. D'Andrea, Luca Domenico et al. "Targeting angiogenesis: structural characterization and biological properties of a de novo engineered VEGF mimicking peptide." *Proceedings of the National Academy of Sciences of the United States of America* vol. 102,40. (2005): 14215-14220. doi:10.1073/pnas.0505047102
4. Shen, Luxuan et al. "Self-adaptive antibacterial scaffold with programmed delivery of osteogenic peptide and lysozyme for infected bone defect treatment." *ACS applied materials & interfaces* vol. 15,1. (2023): 626-637. doi:10.1021/acsami.2c19026
5. Wang, Di et al. "Distinct roles of different fragments of PDCD4 in regulating the metastatic behavior of B16 melanoma cells." *International journal of oncology* vol. 42,5. (2013): 1725-1733. doi:10.3892/ijo.2013.1841
6. Liu, Wenshuai et al. "ECM-mimetic immunomodulatory hydrogel for methicillin-resistant *Staphylococcus aureus*-infected chronic skin wound healing." *Science advances* vol. 8,27. (2022): eabn7006. doi:10.1126/sciadv.abn7006

Thanks again!

Comment 3: How about the surface grafting efficiency of P1 and P2?

Response: Thank you for your thoughtful review and excellent question. To evaluate the grafting efficiency of P1 and P2, titanium plates coated with mussel-inspired peptides were divided into groups of ten and immersed in 0.1 mg/mL solutions of P1 or P2 for 4 hours. The grafting efficiency was quantified by comparing the differences in solution concentrations before and after immersion, as determined by HPLC analysis (SFigure 1).

Figure 1: (A-B) The concentration differences of P1 or P2 solutions before and after grafting were calculated based on the peak areas measured by HPLC. 0.1 mg/mL solution was prepared as the standard prior to grafting, while X1, X2, and X3 represent the concentrations of P1 or P2 solutions measured after grafting.

The grafting efficiency of each titanium plate in the P1 or P2 solution was calculated as follows:

$$E_{P1} = [1 - (503.1 + 502.3 + 506.8) / 3 / 616.8] / 10 * 100\% \approx 1.83\%$$

$$E_{P2} = [1 - (412.0 + 409.7 + 411.9) / 3 / 523.1] / 10 * 100\% \approx 2.14\%$$

The grafting efficiency of P1 and P2 on the titanium surface was relatively low, likely because the 0.1 mg/mL P1 and P2 solution contained a sufficient amount of P1 and P2, facilitating adequate reactions on the titanium surface.

Thanks again!

Comment 4: A previous report showed that the K23 peptide (KAFAKLAARLYRKALARQLGVAA) played its ant-inflammatory role after cell-penetration (doi.org/10.1016/j.nano.2012.09.003). How did the authors guarantee the K23 peptides showed M1 to M2 regulation when the peptides are grafted in the surface of TiO₂?

Response: Thank you for your insightful and valuable comment. The issue you raised has already been considered during the design process of the DOPA-P1@P2 coating. Specifically, K23 was assembled on the outermost layer of the coating and linked to both K15 and Y5 via

the PVGLIG peptide sequence, which can be specifically cleaved by MMP-2 or MMP-9. During LPS-induced macrophage polarization, MMP-2 and MMP-9 secreted by macrophages specifically cleave the PVGLIG peptide sequence, leading to the release of K23 into the culture medium, where it interacts with macrophages to exert its regulatory functions. We further validated in vitro that the PVGLIG peptide sequence can be specifically cleaved by MMP-2, resulting in the successful release of K23 (Figure 1R). To further address your question more clearly, we synthesized CY5.5-labeled K23 and conducted macrophage polarization experiments on the DOPA-P1@P2 titanium surface. The results demonstrated that K23 effectively penetrates macrophages (SFigure 2). These findings suggest that K23 grafted onto the titanium surface can effectively modulate macrophage polarization.

Figure 1 (R) Response and release trends of P1 and P2 following treatment with MMP-2 solution. Treatment with MMP-2 markedly enhances the release of K23 through the specific enzymatic cleavage of the PVGLIG peptide sequence. Within the first 24 hours of MMP-2 treatment, over 60% of K23 was released, followed by a significant slowdown in the release rate after day 3. By the end of the first week, the cumulative release reached approximately 85%.

SFigure 2 Cy5.5-labeled K23 effectively penetrated into BMDMs. Red: Cy5.5-labeled K23; Green: macrophage marker F4/80; Blue: nuclei.

Thanks again!

Comment 5: Did the combination of K23 with K15 or Y5 affect the anti-inflammatory efficiency of K23 peptide?

Response: We greatly appreciate your insightful comment. To address your concern, we performed qRT-PCR experiments to evaluate whether the anti-inflammatory effects of K23 alone differ from those of K23 in combination with K15 (P1) or Y5 (P2) (SFigure 3). The results showed that all three groups significantly downregulated pro-inflammatory genes (*Tnf* and *Il6*) and upregulated the anti-inflammatory gene (*Il10*) compared to the LPS group. However, no significant differences were found among the three groups, suggesting that combining K23 with K15 or Y5 does not alter its anti-inflammatory efficiency.

Figure 3 (A–C) RT-qPCR was used to assess the expression of the pro-inflammatory genes (*Tnf* and *Il6*) the anti-inflammatory gene (*Il10*). The data are presented as the mean \pm standard deviation (SD) (n=3 per group). Statistical analysis was performed via one-way ANOVA (*P < 0.05 and **P < 0.01 vs. the Control group or comparisons between the combined groups indicated in the figure; #P < 0.05 and ##P < 0.01 vs. the LPS group). P < 0.05 indicates statistical significance.

Thanks again!

Comment 6: In Line 178, the author wrote “The DOPA-P1@P2 group presented the smallest water contact 179 angle (55.6°), indicating superior surface hydrophilicity.” As there was no significant difference among DOPA-P1, DOPA-P2, and DOPA-P1@P2 groups, the use of “smallest” was not suitable.

Response: We sincerely appreciate your thorough review. Your valuable comment is crucial for improving the rigor of our manuscript. To ensure accuracy, we have revised the sentence accordingly (**Page 6, Lines 149–151, highlighted in red**).

Revised version:

More precisely, the DOPA-P1@P2 group exhibited a significantly lower water contact angle (55.6°) compared to TiO₂ group (81.1°), indicating superior surface hydrophilicity.

Thanks again!

Comment 7: Which pathway was more important for promoting angiogenesis and osteogenesis, regulation of cytokine secretion by M2 macrophages, or the direct regulation of grafting peptides.

Response: Thank you for your valuable and insightful question. Due to differences in experimental conditions, a direct comparison between the indirect effects (regulation of

cytokine secretion by M2 macrophages) and direct effects (regulation of grafting peptides) on angiogenesis and osteogenesis is not feasible. To address this question, we designated the TiO₂ group under both experimental conditions as the baseline to evaluate the relative improvements in angiogenesis- and osteogenesis-related indicators in the DOPA-P1@P2 group (SFigure 4). The results demonstrated that, under direct intervention conditions, the enhancements in Wound Healing and Number of Junctions (angiogenesis indicators), as well as ALP Positive Area (osteogenesis indicator), were significantly greater than those observed under indirect intervention conditions. These findings suggest that direct effects play a more critical role in promoting angiogenesis and osteogenesis.

SFigure 4 (A–C) Using TiO₂ as the baseline, percentage increases in angiogenesis (Wound Healing, Number of Junctions) and osteogenesis (ALP Positive Area) indicators were observed in the DOPA-P1@P2 group.

Thanks again!

Comment 8: After the transcriptome sequencing analysis, the authors should further prove the selected relative pathway by using RT-PCR, WB or cellular staining.

Response: We greatly appreciate your valuable comment and fully agree with your perspective. We sincerely appreciate your insightful comment and fully agree with your perspective. To address this, we conducted WB analysis to validate the MAPK and PI3K-Akt signaling pathways (Figure S10), which demonstrated consistency between the sequencing

and WB results. Furthermore, we found that the pro-angiogenic effects of DOPA-P1@P2 surfaces may be associated with the activation of extracellular signal-regulated kinase 1/2 (ERK1/2) within the MAPK signaling pathway, while their pro-osteogenic effects are likely linked to the activation of mechanistic target of rapamycin (mTOR) downstream of the PI3K–Akt signaling pathway.

Figure S10: (A–B) WB detected the expression levels of MAPK signalling pathway-related proteins, along with the corresponding quantitative analysis results. (C–D) WB detected the expression levels of PI3K–AKT signalling pathway-related proteins, along with the corresponding quantitative analysis results. The data are presented as the mean ± standard deviation (SD); n = 3 per group. Statistical analysis was performed using Student's t test, and *P < 0.05 and **P < 0.01 indicate statistical significance.

Thanks again!

Comment 9: The authors pointed out the importance of “osteogenic-angiogenic coupling” in Line 320. However, there is no research about the interaction between HUVECs and MSCs.

Response: We greatly appreciate your careful review and constructive feedback on our manuscript. Since this research does not explore the interaction between HUVECs and BMSCs, we have removed the statement on "osteogenic-angiogenic coupling" to maintain the rigor of the manuscript. In future research, the interaction between HUVECs and BMSCs will be a primary focus, and we look forward to presenting the corresponding findings in our upcoming studies.

Thanks again!

Comment 10: The detachment of K23 peptide should be proved *in vivo*. The authors can label K23 peptide with fluorescence.

Response: We sincerely appreciate your insightful and constructive comment. Following your suggestion, we synthesized a CY5.5-labeled K23 peptide and fabricated the DOPA-P1@P2 coating. The unique properties of bone tissue made it difficult to accurately monitor the *in vivo* release of K23 peptide after implanting DOPA-P1@P2-coated titanium rods into the rat femur. This limitation arose primarily from the extended processing time required for decalcification, staining, and imaging of the femoral tissue that often takes at least four weeks, during which the fluorescence signal of CY5.5-labeled K23 could be quenched. To address this issue, we selected the synovial tissue adjacent to the femur as the experimental site, as it is rich in macrophages capable of secreting MMP-2 and MMP-9. These enzymes specifically cleave the PVGLIG sequence, enabling the release of the K23 peptide. Thus, we implanted the trimmed DOPA-P1@P2-coated titanium rods into the synovial membrane of the rat knee joint. This allowed us to examine whether the CY5.5-labeled K23 could respond to inflammation and be released into the synovial tissue within an inflammation-prone microenvironment induced by the presence of a foreign body. At 48 hours post-implantation, synovial tissues were harvested, and the titanium rods were removed for immunofluorescence staining. The results revealed intense red fluorescence signals (CY5.5-labeled K23) within and around macrophages in the synovial tissue (SFigure 5), indicating that K23 could achieve responsive release in the inflammatory microenvironment *in vivo*.

CD68/CY5.5-K23/DAPI

Figure 5 CY5.5-labeled K23 was detected in and around macrophages in the synovial tissue. Red: Cy5.5-labeled K23; Green: macrophage marker CD68; Blue: nuclei.

Thanks again!

Response to Reviewers:

Reviewer #1: The authors have done a thorough job answering this reviewer's concerns, as well as those of the other reviewers, by performing additional studies and more fully characterizing the cell response, both in *vitro* and in *vivo*. The writing is improved. Congratulations on this interesting work.

Response: Thank you for your positive feedback and thoughtful review. We appreciate your valuable insights, which have helped improve our manuscript.

Thanks again!

Reviewer #2: The authors have satisfactorily addressed all comments and suggestions given by all reviewers. I was thinking the same regarding RAW 264.7 cells regarding being prone to polarization, I am glad the authors repeated the experiments on BMDMs. This and the other revisions have significantly increased the quality of the results and support of the hypotheses in the manuscript. I believe the manuscript is now suitable for publication, with one minor suggestion: In the title for figure 2, cytocompatibility is more accurate than biocompatibility. I agree that in vivo experiments shown later demonstrate biocompatibility, but Figure 2 only focusses on in vitro cell culture testing.

Response: We are deeply grateful for your high appreciation of our work. We fully agree with your suggestion and have changed "biocompatibility" to "cytocompatibility" in the title of Figure 2.

Thanks again!

Reviewer #3: The authors have properly addressed all concerns with revisions. It is suggested to be accepted for publication.

Response: Thank you for your supportive feedback and recommendation. We appreciate your time and effort in reviewing our manuscript.

Thanks again!